# Embodied Impact of Facial Coverings: Triggering Self-Expression Needs to Drive Conspicuous Preferences

**DOI:** 10.3390/bs15091150

**Published:** 2025-08-24

**Authors:** Ji Li, Xv Liang

**Affiliations:** Business School, Central University of Finance and Economics, Beijing 100081, China; liji@cufe.edu.cn

**Keywords:** facial covering, embodied cognition, self-expression, conspicuous preferences, face masks

## Abstract

Although prior research has examined how facial covering affects observers’ cognition and attitude, the psychological experiences of individuals with facial coverings themselves and their subsequent behavioral consequences still need to be more explored. From the embodied cognition perspective, we propose facial covering as a direct external stimulus, triggering a psychological gap between the current level of self-expression needs and the diminished self-expression pathways. Using face masks as a specific form of facial covering, five experiments were conducted in China. The results reveal that under facial covering, the surfaced need for self-expression can be transformed into the consumer preference for conspicuousness; and the self-construal type moderates this effect, with independent self-construals exhibiting a stronger covering-induced need for self-expression and subsequent conspicuous preferences compared to interdependent self-construals. The research makes a contribution by enriching the new perspective on the theoretical impact of facial covering. Practically, this research can also provide actionable insights for enterprises in the realms of marketing strategy design and behavior interventions.

## 1. Introduction

The need for self-expression is a fundamental human drive ([5]; [35]), with the intensity of intrinsic motivation varying across individuals ([38]). These driving forces often encourage individuals to actively seek channels to express their inner world. For instance, compared to men, women’s heightened participation in adventure activities and environmental exploration is intrinsically linked to self-discovery and self-expression as core motivators ([26]). Individuals from European and American contexts are more concerned about self-expression than those from East Asian cultures and tend to view speech as an important method of self-expression ([36]). However, the self-expression need is not solely determined by internal factors. Critically, external conditions significantly shape these needs ([78]). Empirical evidence suggests that when people make choices from a wider range of options, they tend to be more inclined towards self-expression ([13]). Compared with a standard appeal, the dueling preferences approach can activate stronger self-expressive motivations ([56]). Collectively, these studies reveal that product selection serves as a means of self-expression, while choice set characteristics shape potential self-expressive need. From this perspective, the human body is an important medium for self-expression, providing a tangible platform for projecting our inner selves ([54]). Yet, a pertinent question that remains inadequately clarified in the existing literature is whether manipulation of the body itself serves as an external stimulus to alter a person’s level of self-expression.

Embodied cognition theory posits that physical sensations or actions can alter perception, highlighting the body’s active role in constructing cognitive experiences ([71]). [80] ([80]) demonstrated that the illusory sense of anonymity induced by sunglasses does not stem from the objective reduction in others’ ability to recognize the wearer due to darkness. Instead, it arises from the wearer’s subjective, phenomenological experience of darkness. This finding reveals that the body’s sensory experiences can directly influence how individuals self-perceive and regulate their behavior. The face is an important biological tool for nonverbal communication ([3]). Facial features reveal stable identity traits, while facial expressivity shows temporary emotions ([49]). Covering the face is an external physical behavior that directly impacts the body’s sensory input. The covering creates a sensation of enclosure and restriction on the facial skin ([20]). When the face is covered, people may feel as though a part of our “self” is being constrained, leading to an imbalance with the current level of self-expression needs. Thus, we infer that facial covering may surface the psychological need for self-expression.

Previous research has mainly focused on how facial covering affects observers, showing, for instance, reduced accuracy in emotion recognition and identity verification ([4]; [8]; [24]), as well as alterations in attitude assessment ([27]; [53]; [77]). Few studies have explored how facial covering influences the wearer’s perception and behavior ([18]). Therefore, our research makes a contribution by enriching the new perspective on the theoretical impact of facial covering. Furthermore, impression management theory suggests that people strategically present themselves to others ([16]; [43]). When channels like faces are partly covered, individuals can redistribute expressive resources to material possessions ([67]). Conspicuous consumption may become a highly relevant alternative. It allows individuals to communicate their self-image to others, perhaps compensating for the reduction of self-expression opportunities caused by facial covering ([14]; [36]). Practically, this research can provide actionable insights for enterprises in the realms of marketing strategy design and behavior interventions.

## 2. Literature Review and Hypotheses Development

### 2.1. Facial Covering and the Need for Self-Expression

The need for self-expression is a fundamental requirement to create and maintain one’s self-identity ([28]; [51]). As social beings, people employ various means to express their personality to the outside world ([7]; [79]). Facial cues provide rich information and are crucial in social interactions ([6]; [63]). When the face is covered, the covering acts as a physical barrier, producing direct and immediate external stimulus. According to embodied cognition, our body’s physical sensations are closely linked to our cognitive and emotional responses ([23]). The sensation of enclosure on the face triggers a psychological gap: the current level of self-expression needs versus the reduction of self-expression pathways. Theoretically, we posit that this gap heightens the need for self-expression.

Hence, the following hypothesis is proposed:

**H1.** 
*Facial covering increases consumers’ need for self-expression.*


### 2.2. Facial Covering: From Self-Expression Gap to Conspicuous Preferences

Individuals have a strong motivation to control the way others perceive them in order to make it consistent with their desired self-concepts ([40]). However, facial covering creates a gap between an individual’s current level of self-expression needs and the reduction of self-expression pathways. To bridge this gap, consumers engage in substitutive self-presentation, reallocating expressive efforts to alternative domains ([65]). Individuals can express their personalities through consumption by reflecting their extended selves via product acquisition ([21]). Material symbols (e.g., clothing and accessories) provide a controllable and visible alternative.

Conspicuous consumption is defined as the act of purchasing and displaying goods primarily to signal social status or wealth ([67]). Conspicuous products are inherently public in nature; their signaling capacity relies on being observed by others ([47]). This not only includes the purchase of high-end products but also encompasses a preference for eye-catching logos ([12]; [22]; [52]; [61]). The symbols employed in conspicuous products are standardized and can be interpreted across different social contexts, requiring minimal cognitive effort from observers for decoding ([29]; [67]). These critical characteristics make conspicuous products a reliable and effective means for individuals to communicate their self-image.

Therefore, under facial covering, the surfaced need for self-expression can be transformed into the consumer preference for conspicuousness. The following hypothesis is proposed:

**H2.** 
*Facial covering has a positive impact on conspicuous preferences. This relationship is mediated by the need for self-expression.*


### 2.3. Moderating Role of Self-Construal

Self-construal is the degree to which individuals define themselves as independent (distinct from others) or interdependent (connected to others) ([46]). This fundamental aspect of self-perception has profound implications for individuals’ self-expression needs and behaviors ([59]).

Individuals with an independent self-construal typically have a strong and inherent need for self-expression ([11]; [72]). They have a cognitive orientation that highly values self-differentiation and the outward display of personal characteristics ([19]; [39]). As a result, they have a relatively high baseline level of self-expression needs, constantly seeking ways to assert their individuality in various situations ([10]; [44]; [50]). In contrast, individuals with an interdependent self-construal have a different set of priorities about self-perception. Their self-concept is more closely tied to their relationships with others and their role within a social group ([62]). They are often taught to prioritize group harmony, conform to social norms, and contribute to the collective well-being ([15]; [17]). This socialization process leads them to rely more on non-individualistic means of self-identification and to place less emphasis on self-differentiation through self-expression ([75]). Consequently, their baseline level of self-expression needs is relatively lower compared to independent self-construal individuals.

When facial covering occurs, it directly disrupts the established self-expression patterns. However, the magnitude of the resulting self-expression need gap differs between independent and interdependent self-construal individuals due to their distinct baseline levels of self-expression needs. We posit that independent self-construal individuals, with their higher baseline needs, will experience a more pronounced self-expression need gap after facial covering. In conjunction with H2, we hypothesize the following:

**H3.** 
*Self-construal moderates the relationship between facial covering and conspicuous preferences. Specifically, individuals with an independent self-construal will experience a heightened need for self-expression after facial covering compared with those with an interdependent self-construal. In turn, this need for self-expression positively impacts conspicuous preferences.*


## 3. Methodology

### 3.1. Hypothesis Validation

Figure 1 illustrates the research framework. To thoroughly test the hypotheses, we carried out five experiments. Study 1 includes Experiment 1 and Experiment 2. Its aim is to verify H1, which states that facial covering increases consumers’ need for self-expression. Study 2 consists of Experiment 3 and Experiment 4. It was designed to check whether the need for self-expression plays a mediating role between facial covering and conspicuous preferences, as H2 suggests. Study 3, through Experiment 5, tests H3, which examines the moderating role of self-construal.

### 3.2. Facial Covering Stimulus Selection

Wearing a mask is an important personal protective measure to reduce the risk of spreading respiratory infectious diseases ([70]). While COVID-19 significantly accelerated the global adoption of mask-wearing, it is essential to note that this practice was already customary in at least some Asian regions prior to the pandemic. For instance, in China, masks have long been used as a safeguard against air pollution ([68]). In Japan, many individuals routinely wore face masks in public settings to prevent the spread of colds or influenza ([48]). [81] ([81]) further elucidated through in-depth interviews that, in the post-pandemic context, functional needs—such as medical defense, protection against fog and dust, warmth and moisture retention, and food and beverage hygiene—as well as emotional needs, including safeguarding facial privacy and expressing unique personalities, collectively contribute to the daily use of face masks.

It should be emphasized that the core of this study is to examine the impact of facial coverings, which is grounded in the real-world perception of physical covering. Consequently, the reasons behind individuals’ decisions to cover their faces are not our focal point. Notably, due to the current diversity in the reasons for mask usage, selecting face masks as the form of facial covering for our research can directly apply to a wide range of commercial scenarios. By doing so, we enhance the practical significance, ensuring that it can effectively address real-world business opportunities.

In addition, face masks come in many types. In this research, we focused on simple surgical masks because they are widely recognized and used, reducing differences in perception due to mask design. Their simplicity keeps the focus on facial covering itself, not on the mask’s features.

### 3.3. Sample Acquisition and Ethical Safeguards

Experiment 1 and Experiment 3 were offline experiments conducted on a university campus in Beijing, China. The experiments took place in a more normal social environment with some participants taking part at the same time. All participants received course credits as compensation. Experiment 2, Experiment 4, and Experiment 5 were online experiments. We used random assignment and sampling from Credamo.com, a well-known Chinese online survey firm with over 3 million members, ensuring sample randomness and representativeness. The consistency of personal characteristics among samples in online experimental settings also proved the stability of our random selection. We randomly grouped participants into those who wore masks and those who did not, balancing potential confounding factors. The online experiments required participants to be in a quiet, undisturbed environment and complete tasks alone to minimize external interference. In the questionnaire, we clearly stated that participants had the right to choose whether to accept this experimental manipulation, and they could exit directly if they could not accept the experimental conditions. Additionally, to avoid response bias, no participant took part in more than one experiment, and they were not informed about the specific objectives related to mask-wearing in the experiment.

Participants were required to be at least 18 years old for adequate cognitive and decision-making capacity, and all were Chinese people living in China to ensure the consistency of norms and values that might influence perceptions and behaviors related to mask-wearing. At recruitment, we excluded participants with several types of medical conditions. Those with respiratory diseases like asthma or bronchitis were not included. These conditions can cause breathing difficulties, and wearing a mask may exacerbate discomfort, distracting participants from experimental tasks and introducing confounding factors unrelated to the embodied effects of facial covering. Participants with facial skin diseases were also excluded. All experiments were conducted subsequent to 2024.

We confirm that all studies were conducted in strict compliance with the Declaration of Helsinki and the ethical standards set by the academic committee of the Business School, Central University of Finance and Economics (protocol code EA20250003). We adhered rigorously to ethical guidelines and took comprehensive measures to protect respondents’ privacy. In the offline experiments, participants were identified solely by their student codes to ensure anonymity. In the online experiments, we explicitly instructed participants to upload a front-facing photograph of themselves that had been processed to remove all sensitive and personally identifiable information, making it impossible to recognize individual features. We clearly explained that the sole purpose of this requirement was to ensure that they adhered to the (with or without) facial covering directive. To safeguard data privacy, access to the data was strictly limited to authorized research team members. After the completion of data analysis, all photographs were permanently deleted from the servers to eliminate any risk of unauthorized use or disclosure.

## 4. Study 1: Examining the Impact of Facial Covering on the Need for Self-Expression

Study 1 aims to preliminarily test H1 through Experiment 1 and Experiment 2. Experiment 1 is a quasi-experiment conducted offline under an experimental design where participants naturally choose whether to wear face masks. Experiment 2 employs more rigorous manipulation, strictly testing the hypothesized relationship by randomly assigning participants to different conditions.

### 4.1. Experiment 1

#### 4.1.1. Participants and Procedure

The experiment was administered in a vacant classroom on campus. The researchers carefully managed the number of participants engaged in the experiment concurrently. In this manner, each participant can be exposed to the social environment without being relatively crowded. The study recruited a total of 95 respondents (55 males and 40 females, M_age_ = 19.44, SD = 0.81, range = 18–22, 100% first-year undergraduate students).

Prior to entering the classroom, participants were notified that masks were freely available at the entrance. The experimenters only recorded the actual status of mask-wearing, where wearing a face mask was coded with a value of 1, and not wearing a mask was coded as 0. To ensure that the participants did not enter the classroom concurrently (which could mitigate the potential influence of the choices of others), the experimenters varied the arrival times for each participant. The data revealed that 42 participants opted to wear masks (24 males and 18 females, M_age_ = 19.50, SD = 0.80, range = 18–22), whereas 53 opted not to wear masks (31 males and 22 females, M_age_ = 19.37, SD = 0.82, range = 18–22).

Given that this experiment measures only a few variables, to lower the chance of participants inferring the true experimental objective, each participant was instructed to mentally simulate a hypothetical shopping scenario. Next, participants completed a paper-based questionnaire to assess their need for self-expression ([28]), using items such as “I have a strong desire for self-expression”, “I want to take actions to let others know who I am”, and “I am motivated to take steps to disclose further information about myself” (7-point Likert scale, 1 = strongly disagree, 7 = strongly agree; Cronbach’s α = 0.899). The participants also provided their gender and age information.

#### 4.1.2. Results

An analysis of variance (ANOVA) revealed a significantly increased need for self-expression among individuals wearing a facial covering (M_with-mask-wearing_ = 4.96, SD = 1.04) compared with those without a facial covering (M_without-mask-wearing_ = 3.75, SD = 1.20; F(1,93) = 26.572, *p* < 0.001), supporting H1.

### 4.2. Experiment 2

#### 4.2.1. Participants and Procedure

A design (face mask-wearing: with vs. without) was employed in Experiment 2. A total of 176 participants (53 males and 123 females, M_age_ = 30.89, SD = 8.67, range = 18–57) were recruited from Credamo.com. Regarding the last finished level of education, 26 had high school education or lower, 111 had a bachelor’s degree, and 39 had a master’s degree or higher.

The participants were randomly assigned to the experimental or control groups. The experimental group was instructed, “Please wear a face mask throughout this experiment”, whereas the control group was instructed, “Do not wear a face mask until the end of the experiment”. After self-checking their readiness to follow instructions, the participants uploaded a front-facing photograph (with sensitive information omitted) of their current state to enable the researchers to ensure that they adhered to the (with or without) mask-wearing directive. The items used to assess the need for self-expression were the same as those in Experiment 1 (Cronbach’s α = 0.750).

Adhering to the manipulation checks employed by [45] ([45]), the participants were asked, “Did you wear a face mask during the experiment?” to reaffirm adherence to the initial instructions. The participants under the with-mask-wearing condition reported wearing masks, whereas those under the without-mask-wearing condition consistently reported not wearing a mask. This correspondence is further corroborated by the manual verification of the photographs, which confirmed a successful manipulation. Lastly, we gathered demographic information from the participants, encompassing gender, age, and education level.

#### 4.2.2. Results

Outcomes of ANOVA revealed a statistically significant increase in the need for self-expression among participants in the experimental group compared with those in the control group (M_with-mask-wearing_ = 5.56, SD = 0.62; M_without-mask-wearing_ = 4.99, SD = 0.97; F(1,174) = 22.491, *p* < 0.001). This finding indicates that facial covering significantly intensified the need for self-expression, supporting H1 again.

### 4.3. Discussion

Using face masks as a specific form of facial covering, both Experiment 1 and Experiment 2 have confirmed that facial covering increases consumers’ need for self-expression. In Experiment 1, participants’ autonomous choice in dividing into the with-mask-wearing and without-mask-wearing groups, without experimenter intervention, minimized the chance of participants guessing the experimental intent and adjusting responses. However, as a quasi-experiment, Experiment 1 may have some unmeasured and unbalanced inter-group differences, introducing bias into causal inferences. This limitation is a common challenge in quasi-experimental research. To mitigate this, Experiment 2 employed stricter controls through random assignment, balancing potential confounding factors and reducing bias in causal reasoning. Using the online platform Credamo.com expanded the sample range, increasing generalizability.

In conclusion, Study 1’s findings set the stage for Study 2, which will further explore the impact of facial covering on conspicuous preferences and the mediating role of the need for self-expression.

## 5. Study 2: Exploring the Influence of Facial Covering on Conspicuous Preferences and the Mediating Role of the Need for Self-Expression

Study 2 was intended to empirically investigate the positive impact of facial covering on conspicuous preferences and to verify the mediating role of the need for self-expression in this relationship. Specifically, Experiment 3 was conducted offline. Apart from testing the aforementioned hypotheses, it also discussed potential interferences such as psychological reactance that might be induced by the forced manipulation of mask-wearing through random assignment ([31]; [60]), and factors like risk perception that could be triggered by the potential association of face masks with diseases or death ([33]; [34]). Experiment 4, conversely, further validated the robustness of the facial covering effect through an online experiment.

### 5.1. Experiment 3

#### 5.1.1. Participants and Procedure

This experiment adopted a single-factor design (face mask-wearing: with vs. without). Similar to the offline setup in Experiment 1, the experiment was carried out in a suitable on-campus location. A total of 67 students (37 males and 30 females, M_age_ = 19.52, SD = 0.84, range = 18–22, 100% first-year undergraduate students) were recruited. Upon arrival, participants were randomly assigned to either the experimental or control group. For the experimental group, participants were provided with clean face masks and instructed to wear them throughout the experiment, while the control group proceeded without any facial covering.

To alleviate potential demand effects stemming from intrusive procedures, we measured the dependent variables first ([32]). The study referred to the experimental design of [57] ([57]) to measure the conspicuous preferences. The participants read a scenario about purchasing a watch. Specifically, Watch A (low conspicuousness) was described as “primarily designed for functionality, serving as an impeccable tool whenever needed”. Conversely, Watch B (high conspicuousness) was characterized as “primarily crafted to create a lasting impression, emphasising one’s status to those in their vicinity”. The participants rated the extent to which they would select between the two watches using a 7-point Likert-type scale (1 = definitely choose Watch A, 7 = definitely choose Watch B).

After completing the product preference task, participants filled out a questionnaire to measure their need for self-expression. The questionnaire used items similar to those in Experiment 1 (Cronbach’s α = 0.927).

In addition, to discuss the potential influence of psychological reactance and risk perception, the measurements were conducted. Six items (Cronbach’s α = 0.881), adapted from [31] ([31]), were employed to assess participants’ psychological reactance in response to being required to wear (or not wear) a face mask during this experiment. These items include the following: “How reasonable do you think this requirement is?”, “How restricted do you feel in your freedom regarding facial covering?”, “How much does it bother you?”, “How irritated do you probably feel?”, “How legitimate is the rule of mandating you to wear (or not wear) a face mask?”, and “How much pressure do you feel to comply with this requirement”. Risk perception was evaluated through nine items such as “I am apprehensive about contracting a respiratory infectious disease” and “I perceive myself as vulnerable to respiratory infectious diseases” ([74]; Cronbach’s α = 0.856). All the above measurements employed a 7-point Likert scale for response. Finally, participants provided their gender and age information.

#### 5.1.2. Results

An ANOVA revealed significantly increased conspicuous preferences among individuals with facial covering (M_with-mask-wearing_ = 5.23, SD = 1.50) compared to those without facial covering (M_without-mask-wearing_ = 2.75, SD = 1.68; F(1,65) = 39.938, *p* < 0.001). However, the study observed no significant differences in psychological reactance (M_with-mask-wearing_ = 4.83, SD = 0.93, M_without-mask-wearing_ = 4.49, SD = 1.08; F(1,65) = 1.946, *p* = 0.168) and risk perception (M_with-mask-wearing_ = 3.08, SD = 0.66, M_without-mask-wearing_ = 3.08, SD = 0.79; F(1,65) = 0.000, *p* = 0.992) between the experimental and control groups. This has to some extent alleviated our concerns about confounding effects. In the mediation analysis, we include these variables as control variables in the model, enhancing the robustness of our analytical results.

The mediating role of the need for self-expression was then examined. Participants with facial covering reported higher levels of need for self-expression (M_with-mask-wearing_ = 5.09, SD = 0.99; M_without-mask-wearing_ = 3.18, SD = 1.00; F(1,65) = 60.981, *p* < 0.001). Subsequently, mediation analysis was conducted using conspicuous preferences as the dependent variable and wearing face masks (0 = without, 1 = with) as the independent variable. The need for self-expression was designated as the mediator, with psychological reactance, risk perception, gender, and age included as control variables. The variables were entered into Model 4 of the PROCESS procedure ([30]; see Figure 2; coefficients have been standardized). The difference in the need for self-expression between the with-mask-wearing and without-mask-wearing conditions (β = 1.385, *p* < 0.001, 95% CI = [1.420, 2.396]) and the difference in conspicuous preferences attributable to the difference in the need for self-expression (β = 0.412, *p* = 0.002, 95% CI = [0.232, 0.974]) were significant and positive.

The results revealed a significant mediation effect of face mask-wearing (with vs. without) on conspicuous preferences (indirect effect (a × b) = 0.571, 95% CI = [0.283, 0.885]) and a significant direct effect (direct effect [c] = 0.657, 95% CI = [0.307, 2.342]), indicating partial mediation by the need for self-expression. Moreover, the proportion of total effect mediated was 46.5%, demonstrating that nearly half of the relationship between facial covering and consumption preferences operates through this psychological mechanism. These metrics underscore the significance of the mediating role played by self-expression needs.

### 5.2. Experiment 4

#### 5.2.1. Participants and Procedure

This experiment continued to employ a single-factor design (face mask-wearing: with vs. without) and was conducted via Credamo.com. The manipulation of the experimental group was consistent with Experiment 2. After random assignment to the experimental (wearing face masks) or control (not wearing face masks) groups, the participants were required to comply with the instructions and upload a photo of themselves (with sensitive information removed). Upon completion, the participants were queried regarding their adherence to the instruction to wear face masks during the survey. One participant failed to comply with the instruction (i.e., they wore a mask despite being instructed not to) and was excluded, which resulted in a final sample of 136 participants (51 males and 85 females, M_age_ = 29.07, SD = 8.54, range = 18–53). Regarding the last finished level of education, 11 had high school education or lower, 107 had a bachelor’s degree, and 18 had a master’s degree or higher.

Following the experimental manipulation, variable measurements were conducted. Regarding the measurement of conspicuous preferences, the logo size was selected as the experimental item ([61]). Specifically, two images of clothing items that differed only in the size of their logos (see Figure 3) were placed side by side and labeled as “A” and “B”. The participants’ preferences for product designs featuring larger (B) and smaller logos (A) were directly assessed using four questions, namely, “Which piece of clothing is more attractive to you?”, “Which piece of clothing would you prefer to wear?”, “Which piece of clothing would you be more willing to spend more money on?”, and “At the moment, which piece of clothing would you choose?”. The items were rated using a 7-point Likert-type scale (1 = definitely choose A, 7 = definitely choose B; Cronbach’s α = 0.985). Higher scores indicate a strong preference for conspicuous products. The items used to assess the need for self-expression were the same as those in Experiment 1 (Cronbach’s α = 0.769). Finally, the participants completed a demographic questionnaire including gender, age, and education level.

#### 5.2.2. Results

The study conducted an ANOVA and set conspicuous preferences and wearing face masks (without = 0, with = 1) as the dependent and independent variables, respectively. The results revealed that the participants wearing a face mask demonstrated a stronger preference for products with higher levels of conspicuousness (M_with-mask-wearing_ = 4.90, SD = 2.43, M_without-mask-wearing_ = 3.80, SD = 2.34; F(1,134) = 7.172, *p* < 0.01).

The mediating role of the need for self-expression was identified. Firstly, participants in the experimental group reported higher levels of need for self-expression (M_with-mask-wearing_ = 5.72, SD = 0.56, M_without-mask-wearing_ = 5.01, SD = 0.89; F(1,134) = 29.984, *p* < 0.001). Then, mediation analysis was conducted using conspicuous preferences as the dependent variable and wearing face masks (0 = without, 1 = with) as the independent variable. The need for self-expression was designated as the mediator, with gender, age, and education level included as control variables. The variables were entered into Model 4 of the PROCESS procedure ([30]; see Figure 4; coefficients have been standardized). The difference in the need for self-expression between the with-mask-wearing and without-mask-wearing conditions (β = 0.852, *p* < 0.001, 95% CI = [0.448, 0.956]) and the difference in conspicuous preferences attributable to the difference in the need for self-expression (β = 0.209, *p* < 0.05, 95% CI = [0.074, 1.028]) were significant and positive. The results revealed a significant mediation effect of face mask-wearing (with vs. without) on conspicuous preferences (indirect effect (a × b) = 0.178, 95% CI = [0.035, 0.347]) and a significant direct effect (direct effect [c] = 0.287, 95% CI = [−0.160, 1.406]), indicating full mediation by the need for self-expression in this experimental scenario.

### 5.3. Discussion

Study 2 achieved a transition from examining behavioral impacts on psychology to investigating psychological mechanisms influencing behavior, thereby elucidating the causal pathway through which facial covering affects conspicuous preferences. By empirically validating the mediating role of the need for self-expression across multiple experimental contexts, this study significantly enhances the robustness of our theoretical framework.

Notably, Experiment 3 took into account potential confounding factors, including psychological reactance and risk perception. Our results show that the mask-wearing manipulation may not have substantially activated notable cognitive differences related to these factors. We posit that this outcome is likely attributable to the broad social acceptance of mask use in the post-COVID-19 era. The efficacy of masks has been widely recognized by the public, and the act of wearing masks has transitioned from being a special measure to a daily routine. This has essentially minimized the association with freedom restrictions or disease-related anxiety. Given that social norms and values related to mask-wearing may vary across different countries or regions, we circumspectly state that this conclusion can be applied to regions with a socio-cultural environment similar to that of China.

A particular finding emerges from the differential mediating patterns observed across experimental modalities. In the (offline) Experiment 3, the need for self-expression partly mediated the relationship, while (online) Experiment 4 revealed a full mediation (the direct effect was not significant). It likely reflects the complexity of real-world interactions. Compared to online settings, physical laboratory spaces introduce uncontrolled variables such as ambient noise, incidental social observations, and interpersonal proximity. These factors may activate other psychological or behavioral mechanisms beyond self-expression needs. Experiment 4 provided a more isolated testing environment, which allowed for a clearer examination of the mediating mechanism. However, in Experiment 4, we found that the direct effect was numerically larger than the indirect effect. This suggests that we cannot completely neglect the existence of other paths. Therefore, this study cautiously posits that the need for self-expression plays a partial mediating role in the relationship between facial covering and conspicuous preferences.

In the following study, we explored how individual differences in self-construal further moderate this mechanism.

## 6. Study 3: Testing the Moderating Role of Self-Construal

Experiment 5 investigated the moderating effect of self-construal on the influence of facial covering on conspicuous preferences. Employing a 2 (face mask-wearing: with vs. without) × 2 (self-construal: independent vs. interdependent) design, this experiment recruited 120 participants (46 males and 74 females, M_age_ = 31.96, SD = 8.55, range = 18–55) from Credamo.com. Regarding the last finished level of education, 29 had high school education or lower, 72 had a bachelor’s degree, and 19 had a master’s degree or higher.

### 6.1. Procedures and Measures

The experiment was conducted in three parts. Firstly, the study manipulated and measured participants’ self-construal. The participants were instructed to read materials that primed either an independent (or interdependent) self-construal and then copy sentences depicting their psychological activities as the protagonist ([73]). Afterward, they completed the self-construal scale ([55]; 7-point Likert-type scale [1 = strongly disagree, 7 = strongly agree] with 6 items for independent self-construal [Cronbach’s α = 0.826] and 10 items for interdependent self-construal [Cronbach’s α = 0.905]).

Secondly, a manipulation of wearing face masks (or not) was implemented. Similar to Experiment 2, the participants were randomly assigned to the with-mask-wearing or without-mask-wearing groups and completed the corresponding instructions.

Thirdly, variable measurements were conducted. To assess the conspicuous preferences, participants were instructed to imagine shopping for high-end designer clothing in a mall ([41]). The conspicuous preference was reflected through responses to four items regarding the visibility and prominence of brand logos (e.g., very invisible/visible) using a 7-point Likert-type scale (Cronbach’s α = 0.919). High scores indicate high levels of preference for conspicuousness. The measurement method for the mediator (need for self-expression) was the same as in Experiment 1 (Cronbach’s α = 0.851). Finally, the participants provided demographic information, including gender, age, and education level.

### 6.2. Results

For manipulation checks, the self-construal index, which was computed following [76] ([76]), indicates a stronger tendency towards independent self-construal with high values and a strong tendency towards interdependent self-construal with low values. Subsequently, the ANOVA results revealed that the self-construal index for the group primed with independent self-construal (M = 0.78, SD = 1.52) was significantly higher than that of the group primed with interdependent self-construal (M = −0.38, SD = 1.51), F(1,118) = 17.435, *p* < 0.001), confirming successful manipulation.

Using Model 7 in PROCESS ([30]; assuming that the first half of the mediation model is moderated), the study verified the moderating role of self-construal. Face mask-wearing (without = 0, with = 1) and need for self-expression were set as the independent and mediator variables, respectively. The type of self-construal (0 = interdependent, 1 = independent) was used as the moderator, and gender, age, and education level were considered as control variables. Lastly, the conspicuous preference was entered as the dependent variable. The results (Figure 5) indicate a significant interaction between wearing face masks and self-construal (β = 0.654, t = 1.778, *p* = 0.078). Specifically, wearing face masks exerted a significant positive effect on the need for self-expression among individuals with an independent self-construal (95% CI: LLCT = 0.272, ULCI = 1.294), with an effect size of 0.783. In contrast, this effect was non-significant among individuals with an interdependent self-construal (95% CI: LLCT = −0.390, ULCI = 0.649). Furthermore, the need for self-expression (mediator) positively impacted conspicuous preferences (β = 0.623, t = 6.101, *p* < 0.001). These findings provide support to H3, demonstrating that compared with individuals with an interdependent self-construal, those with an independent self-construal exhibit higher levels of need for self-expression when using a facial covering, which subsequently positively impacted their conspicuous preferences.

### 6.3. Discussion

Study 3 demonstrates that self-construal serves as a moderator in the facial covering effect, shaping how individuals perceive and respond to environmental constraints on self-expression. In this study, we focused on individual-level self-construal and ensured, through experimental control, that participants were from relatively similar social and cultural backgrounds. This allowed us to isolate the impact of individual self-construal on the psychological responses to facial covering. Facial covering triggers a psychological gap between the current level of self-expression needs and the reduction of self-expression pathways. Independent self-construal individuals, with their higher baseline needs, will be significantly affected. However, interdependent self-construal individuals have lower baseline needs. The reduction in facial visibility does not create as large a gap in their self-expression level. As a result, this leads to pronounced differences in how different individuals behave under the effect of facial covering.

It is important to note that when considering a broader, cross-cultural context, the influence of macro-cultural factors, such as collectivism and individualism, on the development of self-construal must also be taken into account. This highlights the multi-layered nature of the relationship between environmental constraints, self-construal, and psychological responses.

## 7. Discussion and Conclusions

Using face masks as a specific form of facial covering, this study explores the psychological and behavioral repercussions. Drawing on the theory of embodied cognition, we propose that facial covering acts as a physical barrier to natural self-expression, thereby exacerbating the imbalance between the current level of self-expression needs and the reduction of available self-expression pathways. In the context of consumption, this revealed self-expression need motivates consumers to seek highly recognizable material symbols, subsequently heightening conspicuous preferences. We also examine the moderating role of self-construal, demonstrating that independent self-construal individuals exhibit stronger covering-induced need for self-expression and subsequent conspicuous preferences compared to the interdependent self-construals.

### 7.1. Theoretical Implications

First, this study deepens the understanding of how facial covering influences the consumption behaviors of those who are face-covered. Unlike existing research that primarily emphasizes how facial covering affects observers’ perception ([1]; [9]; [42]; [64]), our research explores the psychological changes and subsequent behavioral manifestations experienced by individuals with some facial covering. We found that the need for self-expression plays a mediating role between facial covering and conspicuous preferences. This adds to the literature on behaviorally triggered psychological changes in consumption, offering new insights into how environmental constraints on self-expression can drive specific consumption patterns. By verifying the moderating role of self-construal, we have a better understanding of consumer heterogeneity and the nuanced ways in which cultural and psychological dispositions shape responses to expressive disruptions. These findings provide a basis for future research to further explore the complex relationship.

Second, we focus on the transformation of the need for self-expression into conspicuous preferences. Seeking uniqueness, making creative purchases, or achieving self-brand congruence all appear to serve as means of self-concept compensation ([37]; [66]; [82]). However, our study reveals a distinct pathway. The key reason lies in the characteristics of conspicuous consumption: public visibility and symbolic standardization. The stable semantic connotations and easily interpretable characteristics of conspicuous product symbols make them a more reliable and effective means for individuals to communicate their self-image. In contrast, regarding the product itself, individuals hold divergent perspectives on whether a given item exhibits uniqueness ([25]). Similarly, creative purchases, while reflecting the individual’s creativity, do not have the same broad and consistent social recognition as conspicuous products ([58]). As for self-brand congruence, it focuses more on the alignment between an individual’s self-concept and the brand image, which is an internal form of identification. This concept itself does not emphasize the outward transmission of signals ([2]). Therefore, by validating the pathway from the need for self-expression to conspicuous preferences under facial covering, our study offers more specific insights into the relationship between self-expression needs and consumption behaviors.

### 7.2. Practical Implications

Our study utilizes face masks as a form of facial covering and demonstrates through experiments that the facial covering effect persists regardless of whether individuals actively choose to wear masks or not. This finding provides a solid foundation for businesses to develop targeted marketing strategies and conduct effective consumer behavior interventions.

In offline settings where mask-wearing has become commonplace and during times such as summer and winter when people often wear masks, stores can strategically place conspicuous products at the front of the store or in high-traffic areas to attract consumers’ attention. Additionally, sales staff can be trained to highlight the conspicuous features of products, such as large logos or promotional slogans with conspicuous attributes, when interacting with customers. By doing so, businesses can tap into consumers’ heightened need for self-expression, which is triggered by facial covering, and drive sales of conspicuous products.

In online environments, e-commerce platforms can analyze user behavior data, such as browsing history and purchase patterns, combined with geographic location information to estimate the likelihood of people wearing masks in different areas or during specific times. Based on this information, platforms can precisely target and promote highly conspicuous products to consumers in areas or at times with high mask usage. For example, during peak commuting hours when many users are likely to be wearing masks on public transportation ([69]), online retailers can increase the visibility of conspicuous products in their digital advertisements and product recommendations displayed on mobile apps or websites. This approach allows businesses to meet consumers’ need for self-expression, thereby facilitating product sales.

In addition, considering product characteristics, businesses have the ability to shape consumers’ self-construal types through advertisements and interactive games. By doing so, they can manage and adjust how the wearing of face masks affects consumers’ conspicuous preferences. When products exhibit high levels of conspicuousness, activating the independent self-construal of consumers is advisable; conversely, for products with low levels of conspicuousness, initiating the interdependent self-construal may be more beneficial.

### 7.3. Limitations and Directions for Future Research

While this study provides theoretical and practical contributions, it also has its limitations. Specifically, future research could advance in the following directions.

One limitation is the potential restriction of sample generalizability. The research was conducted within a Chinese cultural context. Given that social norms, values, and behaviors related to mask-wearing and self-expression may vary across different countries or regions, the conclusions of our current study can be applied to regions with a socio-cultural environment similar to that of China. For instance, due to the common acceptance of face masks in China, we did not specifically take into account the impact of individuals’ willingness to wear masks in our research design. Additionally, although psychological resistance did not seem to significantly affect our research results in this study, in regions where mask-wearing is less socially accepted or in more individualistic cultures, its influence could be stronger ([31]). This may alter the underlying mechanism or affect result replication. Future research should replicate our study in diverse cultural settings to assess the cross-cultural validity of our findings and explore potential variables that may influence the strength or direction of these relationships.

While our study focused on face masks as the form of facial covering, we think that other types of attire, such as burqas, may also induce a sense of physical restriction on the facial skin, potentially triggering similar psychological and behavioral responses. However, different coverings inherently carry distinct connotations and cover different body parts. This may lead to variations in their effects. Future research ought to investigate and further substantiate the applicability of a wider array of facial covering forms, thereby augmenting the contributions of our research. Moreover, conspicuous preferences represent merely a self-expression pathway within the consumption realm, which may be achieved through other means in different contexts. Therefore, future studies could delve deeper into this aspect for further exploration.

Unlike many prior studies that directly focus on the psychological effects on behavior, we delve into how a behavioral change (facial covering) triggers a psychological need (for self-expression), which in turn influences consumption preferences. However, despite our innovative focus on this causal chain, it is important to note that while our independent variable (facial covering) was manipulated through behavior, our other variables still relied on self-report measures. Self-report methodologies, though widely used for capturing subjective experiences, are inherently prone to social desirability bias. To reduce this limitation, future research could employ physiological indicators or conduct behavioral observations.

## Figures and Tables

**Figure 1 behavsci-15-01150-f001:**
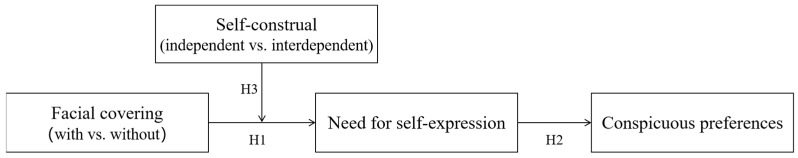
Research framework.

**Figure 2 behavsci-15-01150-f002:**
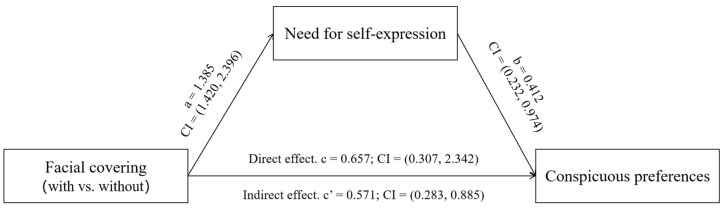
Path analysis of variables.

**Figure 3 behavsci-15-01150-f003:**
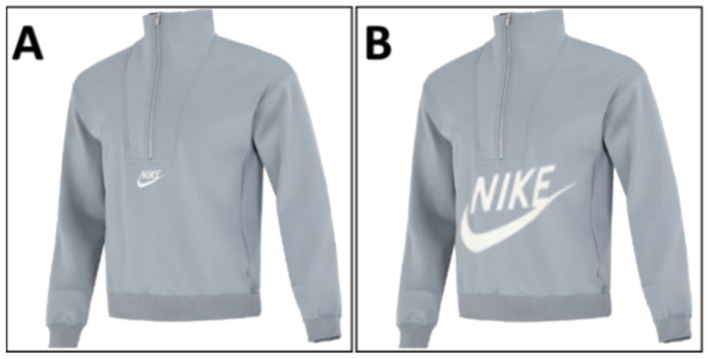
Experimental materials for Experiment 4: (**A**) Clothing item with a smaller logo; (**B**) Clothing item with a larger logo.

**Figure 4 behavsci-15-01150-f004:**
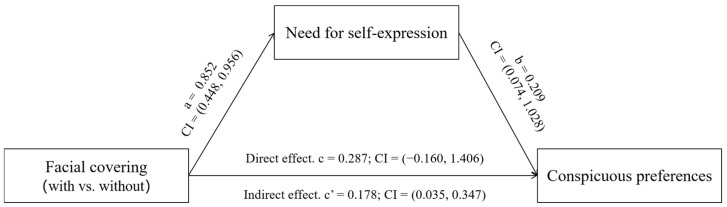
Path analysis of variables.

**Figure 5 behavsci-15-01150-f005:**
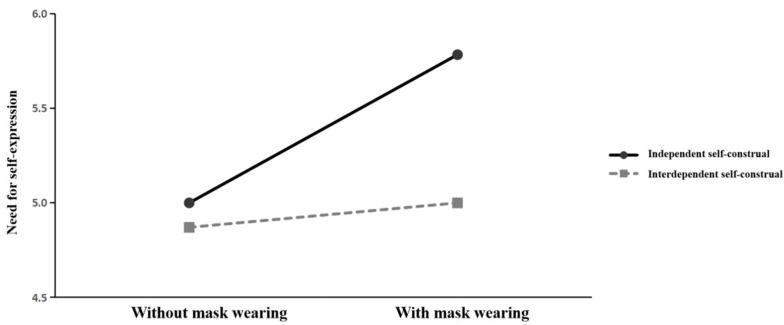
Moderating effect of self-construal.

## Data Availability

The data are not publicly available due to containing information that may compromise the participants’ privacy.

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
