# Peer review of "Embodied Impact of Facial Coverings: Triggering Self-Expression Needs to Drive Conspicuous Preferences"

_behavsci, 2025, doi:10.3390/bs15091150_

Round 1

Reviewer 1 Report

Comments and Suggestions for Authors

Reading this manuscript, it feels like a good product in the wrong packaging. I can see the potential of this article. The research topic is sound. However, I would suggest that the authors rewrite this paper. It would be better for the authors to change the angle and focus of the paper to emphasise more on the consumer psychology concepts.

The title should highlight the concepts, instead of the context (face mask), because it sounds a bit outdated (post-COVID, face masks are no longer compulsory). In the introduction, the mention of cosmetics sales is misleading to readers, as the context and contribution of this work are not about cosmetics.

The flow of the literature review needs improvement to enhance the clarity of this article's focus. The literature review should highlight the consumer psychology concepts more, rather than the context (wearing a face mask). The authors can consider explaining the setting in the methodology.

The methodology can start with the context scenario for the experiment. Here, the authors can mention the face mask scenario, explain the sampling, experiment manipulation, and so on.

The discussions should highlight the conceptual findings and link them with the literature. Accordingly, the authors can emphasise more theoretical and practical contributions.

Comments on the Quality of English Language

The authors need to improve the flow and structure of this manuscript.

Author Response

Comment 1: Reading this manuscript, it feels like a good product in the wrong packaging. I can see the potential of this article. The research topic is sound. However, I would suggest that the authors rewrite this paper. It would be better for the authors to change the angle and focus of the paper to emphasise more on the consumer psychology concepts.

Response 1: Thank you for making us realize the potential of this article. As you have pointed out, masks are just a means of manipulation, and our focus should be on the fundamental concepts they embody. So, we have made comprehensive adjustments from the title and abstract to the introduction, hypothesis derivation, experimental methods, discussion, and conclusion sections to ensure that the entire paper consistently highlights the core concepts and their relationships. All the modifications are highlighted in yellow in the manuscript. 

Now, please review these responses to your specific comments below.

Comment 2: The title should highlight the concepts, instead of the context (face mask), because it sounds a bit outdated (post-COVID, face masks are no longer compulsory). In the introduction, the mention of cosmetics sales is misleading to readers, as the context and contribution of this work are not about cosmetics.

Response 2: We sincerely appreciate your attention to the novelty of our research.

In the post-COVID-19 era, masks are no longer mandatory. Focusing only on masks will greatly limit the theoretical scope of our research. Therefore, we define the more fundamental concept of masks as a specific form of facial occlusion. We consistently applied the concept of facial occlusion and reconstructed the research logic accordingly.

We have updated the title to “Facial Occlusion Triggers the Need for Self-Expression to Drive Conspicuous Preferences”. Based on this change, we have also removed misleading references to cosmetics sales in the introduction, as you pointed out, which does not align with the core background and contribution of our work.

Thank you again for your valuable feedback, which will help us improve the manuscript and better convey the essence of our research.

Comment 3: The flow of the literature review needs improvement to enhance the clarity of this article's focus. The literature review should highlight the consumer psychology concepts more, rather than the context (wearing a face mask). The authors can consider explaining the setting in the methodology.

Response 3: We sincerely appreciate your thoughtful guidance. Literature reviews should place greater emphasis on consumer psychology concepts. Therefore, we have made significant adjustments.

We assume that the core theme of the derivation is the relationship between facial occlusion and self-expression needs, as well as how self-expression needs under facial occlusion conditions lead to conspicuous preferences. We deliberately avoid using the concept of masks in the hypothesis derivation process to ensure a more conceptual and theoretically reasonable discussion.

As you suggested, we clearly specify the masks in Section “3. Methodology”. We also explained in detail the reasons for choosing masks. This arrangement helps readers understand the connection between experimental methods and the core concepts we are studying.

We sincerely appreciate your suggestion to standardize our hypothesis derivation and enhance the overall logical coherence.

Comment 4: The methodology can start with the context scenario for the experiment. Here, the authors can mention the face mask scenario, explain the sampling, experiment manipulation, and so on.

Response 4: Thank you for your detailed guidance. We have thoroughly revised “3. Methodology”.

Firstly, we explained the choice of masks as a form of facial occlusion, as they are a commonly used and widespread means of creating facial occlusion conditions in real life. We also elaborated on how to protect the privacy of participants and the source of experimental samples. In addition, in sections 4, 5, and 6, we explained how to manipulate the mask in each experiment. This detailed description enables readers to accurately understand the experimental procedure and improves the transparency.

We believe that your suggestion has greatly improved the clarity of our research.

Comment 5: The discussions should highlight the conceptual findings and link them with the literature. Accordingly, the authors can emphasise more theoretical and practical contributions.

Response 5: We sincerely appreciate your insightful guidance. We fully recognize the importance of discussing concepts to enhance the theoretical depth and value of our research.

We completely rewrote “7. Discussion and Conclusion”, focusing on the relationship between facial occlusion, the need for self-expression, and conspicuous preferences. In addition, we also elaborated on the theoretical and practical contributions of our research.

Theoretical contributions

Our research provides several theoretical insights. Unlike previous studies that mainly focused on the observer's perception of facial occlusion, we delved into the psychological and behavioral changes of the occluded individuals themselves. By identifying the need for self-expression as a mediating role between facial occlusion and conspicuous preferences, we have added a way to understanding the consumption related psychological changes triggered by behavioral factors. In addition, we found that the need for self-expression under facial occlusion can be transformed into conspicuous preferences, offering more specific insights into the relationship between self-expression barriers and consumption behaviors. The verification of the moderating effect of self-construal has also deepened our understanding of consumer heterogeneity.

Practical Contributions

We have conducted in-depth discussions on how to use research findings to guide marketing practices in three major areas, including online and offline scenarios, and intervention methods for enterprises. In addition, on “7.3 Limitations and Directions for Future Research”, we suggest that future research could explore other forms of facial occlusion, such as burqas or sunglasses, which may also become barriers to self-expression and trigger similar psychological and behavioral responses. This suggestion aims to further expand the practical value of our research.

Thank you again for your valuable suggestion, which greatly improves the quality and significance of our manuscript. 

Reviewer 2 Report

Comments and Suggestions for Authors

This manuscript explores an original and timely topic.

The theoretical grounding is solid, and the manuscript demonstrates a commendable effort to integrate self-expression theory, self-construal, and conspicuous consumption within a coherent analytical framework. However, there are several areas where the manuscript could be strengthened:

Clarify and deepen theoretical positioning (e.g., page 2, lines 73–92):
While the literature review introduces key concepts effectively, the link between face mask-induced self-expression deprivation and compensatory consumption could benefit from greater theoretical elaboration. Specifically, expanding on why mask-wearing inhibits self-expression beyond facial cues (e.g., non-verbal communication or voice muffling) could strengthen the rationale. Including a brief discussion of related work in embodied cognition or impression management would also enhance depth.

Improve precision in hypothesis development (e.g., page 3, lines 124–127):
The hypotheses are clearly stated but could be more sharply framed. For instance, H2 would benefit from a brief justification of why conspicuousness—rather than, say, uniqueness or self-narrative consumption—is the predicted compensatory strategy. Clarifying this would better align the reader’s expectations with the experimental design.

Strengthen methodological transparency (e.g., page 5, lines 165–186):
The description of the experimental manipulation involving mask-wearing is adequate, but more detail on the verification process (photo analysis criteria, data privacy measures) would enhance credibility and replicability. Furthermore, although participant recruitment is described as random, additional information about stratification, balancing across demographic groups, or exclusion criteria should be provided.

Refine data interpretation and presentation (e.g., page 6, lines 229–245):
The mediation analysis is compelling, but the interpretation would be clearer if the authors visualized the indirect effect using a path diagram (e.g., with standardized coefficients). Additionally, while the statistical significance is emphasized, effect sizes should be discussed more explicitly in practical terms.

Clarify moderation analysis logic (e.g., page 7, lines 273–296):
The moderating role of self-construal is an interesting and valuable contribution. However, the text could benefit from a more intuitive explanation of why individuals with interdependent self-construal did not show the same compensatory response. Cultural context may be relevant here and should be briefly explored.

Enhance the conclusion and implications (e.g., pages 8–9):
The conclusion section is generally strong, but its practical implications could be more concrete. For example, suggesting how marketers could use these insights in digital advertising (e.g., tailoring product visibility based on mask mandates or regional norms) would improve applicability. Additionally, the limitations section could acknowledge the potential for social desirability bias in self-report measures, particularly regarding mask compliance.

Minor formatting and referencing issues:
Ensure consistent reference formatting (some URLs appear broken across lines) and verify that all in-text citations (e.g., “Lee and Lee, 2024”) are properly listed in the reference section.

Author Response

Comment 1: This manuscript explores an original and timely topic.

The theoretical grounding is solid, and the manuscript demonstrates a commendable effort to integrate self-expression theory, self-construal, and conspicuous consumption within a coherent analytical framework. However, there are several areas where the manuscript could be strengthened:

Response 1: Thank you very much for your positive feedback on our manuscript. We have taken note of the specific guidance you provided below and incorporated it into the revision of this manuscript.

Comment 2: Clarify and deepen theoretical positioning (e.g., page 2, lines 73–92):

While the literature review introduces key concepts effectively, the link between face mask-induced self-expression deprivation and compensatory consumption could benefit from greater theoretical elaboration. Specifically, expanding on why mask-wearing inhibits self-expression beyond facial cues (e.g., non-verbal communication or voice muffling) could strengthen the rationale. Including a brief discussion of related work in embodied cognition or impression management would also enhance depth.

Response 2: Thank you for your specific suggestions on enhancing our study's theoretical foundation. We've revised the manuscript accordingly.

We have refined the theoretical framework's logic. Starting with the biological basis of facial features as a key tool for nonverbal communication, we emphasize that facial features convey both stable identity traits and immediate emotions through expressions, making the face an important channel for self-expression. Thus, facial coverings like masks disrupt this channel, thereby triggering subsequent psychological needs.

For theoretical depth, we've centered on impression management theory. This theory highlights that individuals are motivated to control how they're perceived to match their desired self-image. Facial occlusion pushes individuals to shift their expressive focus to material symbols like clothing. We further explain how conspicuous consumption, with its public visibility and symbolic standardization, becomes an ideal substitute for self-expression when faces are covered. Its high visibility ensures messages are sent, while standardized symbols simplify interpretation, effectively fulfilling expression needs.

The above modifications include “1. Introduction” and “2. Literature Review and Hypotheses Development” (page 1, line 26 - page 3, line 125). For detailed information, please refer to the highlighted text in yellow.

Thank you again for guiding us in this crucial direction.

Comment 3: Improve precision in hypothesis development (e.g., page 3, lines 124–127):

The hypotheses are clearly stated but could be more sharply framed. For instance, H2 would benefit from a brief justification of why conspicuousness—rather than, say, uniqueness or self-narrative consumption—is the predicted compensatory strategy. Clarifying this would better align the reader’s expectations with the experimental design.

Response 3: We strongly agree on the importance of clearly stating assumptions. Especially the logical process you pointed out from self-expression needs to conspicuous preferences when covering the face. Providing a strong explanation for why conspicuous consumption not only enhances the persuasiveness of our research framework, but also elucidates the unique contribution of our study.

Based on your feedback, we have implemented the following revisions:

In the section "2. Literature Review and Hypothesis Development" (page 2, line 61 - page 3, line 125), we thoroughly reconstructed the logical derivation process. We have now clearly explained how the disruption of self-expression caused by facial occlusion leads individuals to seek alternative ways of expression, and how conspicuous consumption, with its high public visibility and symbolic standardization, is particularly suitable for meeting this need.

In the second paragraph of Section "7.1 Theoretical Implications" (page 12, line 513 - page 12, line 530), we specifically discussed distinguishing conspicuous consumption from other potential compensation strategies. By emphasizing the specific attributes and advantages of conspicuous consumption, our goal is to clearly demonstrate its uniqueness and strengthen its central position in our research.

Thank you again for your constructive feedback, which will help improve our research.

Comment 4: Strengthen methodological transparency (e.g., page 5, lines 165–186):

The description of the experimental manipulation involving mask-wearing is adequate, but more detail on the verification process (photo analysis criteria, data privacy measures) would enhance credibility and replicability. Furthermore, although participant recruitment is described as random, additional information about stratification, balancing across demographic groups, or exclusion criteria should be provided.

Response 4: Thank you for your suggestion on strengthening the disclosure of experimental details.

In the revised manuscript, “3. Methodology” (page 3, line 126 - page 4, line 172) provides a comprehensive introduction to research settings and experimental objectives. We have clearly explained why masks were chosen as the facial occlusion method for our investigation, and provided detailed information on the sample sources and relevant ethical statements for each experiment. In the fourth paragraph, we described the target of photo collection in the online experiment, the program for photo processing, and the data privacy protection measures we took.

In addition, in the experimental descriptions of sections 4-6, we disclose demographic data of the sample, such as gender, age, and education level. These variables are crucial for understanding the composition of the sample and ensuring its representativeness. In the mechanism testing, we used these demographic data as control variables to minimize the potential impact of confounding factors related to demographic characteristics, thereby improving the internal validity of our results.

Comment 5: Refine data interpretation and presentation (e.g., page 6, lines 229–245):

The mediation analysis is compelling, but the interpretation would be clearer if the authors visualized the indirect effect using a path diagram (e.g., with standardized coefficients). Additionally, while the statistical significance is emphasized, effect sizes should be discussed more explicitly in practical terms.

Response 5: Thank you for your specific guidance on data interpretation and presentation.

We already added a path diagram for visualization in “5. Study 2: Exploring the Influence of Facial Occlusion on Conspicuous Preferences and the Mediating Role of the Need for Self-Expression” (page 8, line 325; page 9, line 382). We also discussed effect sizes in the data results (page 7, line 310 - page 7, line 324). In addition, we used “5.3 Discussion” section (page 9, line 384 - page 10, line 414) provides a comprehensive discussion.

Comment 6: Clarify moderation analysis logic (e.g., page 7, lines 273–296):

The moderating role of self-construal is an interesting and valuable contribution. However, the text could benefit from a more intuitive explanation of why individuals with interdependent self-construal did not show the same compensatory response. Cultural context may be relevant here and should be briefly explored.

Response 6: Thank you for your attention and guidance on moderating variables. Based on your suggestion, we have focused on revising the hypothesis derivation and empirical result discussion in the manuscript.

In section “2.3. Moderating Role of Self-Construal” (page 3, line 97 - page 3, line 125), we point out that individuals with an interdependent self-construal derive self-concept from social harmony, and group inclusion. Facial occlusion poses less of a threat to their social identity, as interdependents rely on shared norms and relational networks rather than fixed personal attributes to maintain status. Moreover, interdependents are sensitive to normative pressures. Conspicuous consumption, which risks violating norms of modesty in collectivist cultures, is thus less appealing as a compensatory strategy.

In the section "6. Study 3: Testing the Moderating Role of Self-Construal", we have added a new "6.3 Discussion" (page 11, line 469 - page 12, line 485) to discuss the relevant content of culture and individual level self-construal.

These modifications will deepen the understanding of the moderating role of self-construal.

Comment 7: Enhance the conclusion and implications (e.g., pages 8–9):

The conclusion section is generally strong, but its practical implications could be more concrete. For example, suggesting how marketers could use these insights in digital advertising (e.g., tailoring product visibility based on mask mandates or regional norms) would improve applicability. Additionally, the limitations section could acknowledge the potential for social desirability bias in self-report measures, particularly regarding mask compliance.

Response 7: We fully agree with your viewpoint that enhancing the conclusion and implications is of great significance. We have made corresponding modifications to the manuscript.

In section “7.2. Practical Implications” (page 12, line 531 - page 13, line 561), we have conducted in-depth discussions on how to use research findings to guide marketing practices in three major areas, including online and offline scenarios, and intervention methods for enterprises.

In the fourth paragraph of "7.3 Limitations and Directions for Future Research" (page 14, line 580 - page 14, line 588), we specifically discussed the issue of the potential for social desirability bias in self-report measures.

Thank you again for your valuable insights.

Comment 8: Minor formatting and referencing issues:

Ensure consistent reference formatting (some URLs appear broken across lines) and verify that all in-text citations (e.g., “Lee and Lee, 2024”) are properly listed in the reference section.

Response 8: Thank you very much for bringing this matter to our attention. In this revision, we have thoroughly revised all references. We've paid special attention to the issues you pointed out regarding URLs and the alignment between in-text citations and the reference. We sincerely appreciate your guidance.

Reviewer 3 Report

Comments and Suggestions for Authors

The paper looks at an interesting consequence of mask-wearing in the post-pandemic world. Specifically, how mask-wearing, normalized during the COVID-19 pandemic, influences consumers’ need for self-expression and subsequent preferences for conspicuous consumption. Drawing on compensatory consumption processes, the authors propose that mask-wearing limits typical self-expressive outlets (e.g., facial cues), thereby increasing the need for alternative forms of expression through product choices. Across three studies, they demonstrate that mask-wearing heightens the desire for self-expression, which in turn increases preferences for conspicuous products (e.g., items with larger brand logos). This effect appears contingent upon a consumer’s self-construal (i.e., independent vs. interdependent), such that those with independent self-construals show heightened conspicuous self-expression preferences. The paper situates itself nicely within the literature on consumer identity and self-expression, and aims to offer insights into how individual differences shape consumer behavior in the post-pandemic era.

One thing I struggled with in reading your paper lies in the leap from self-expression to conspicuous consumption, particularly when considered through the lens of self-construal. The authors argue that individuals with an independent self-construal, who prioritize autonomy and distinctiveness, are more likely to engage in conspicuous consumption when mask-wearing restricts facial self-expression. However, it is unclear why conspicuous products, such as a Nike sweatshirt with a large logo, would serve as an effective outlet for individual self-expression, given that such items are often mass-produced and socially conforming rather than unique or identity-revealing. This raises the question of whether the observed behavior reflects a compensatory response to a deficit in self-expression, or instead a reactive response to a perceived restriction on autonomy (e.g., reactance). Independent self-construals are more sensitive to constraints, which makes them more susceptible to psychological reactance (Sittenthaler et al., 2015). From this perspective, the preference for conspicuous products may reflect an effort to reassert agency or visibility in the face of externally imposed limitations, rather than to compensate for an expressive deficit. A clearer articulation of why conspicuousness, rather than uniqueness, creativity, or self-brand congruence, serves as the primary expressive outlet would strengthen the results and takeaways from the paper. Additionally, considering or ruling out reactance as an alternative explanation would help clarify the underlying motivational mechanism, particularly given that the self-construal moderation pattern may align more closely with a reactance framework.

Another potential mechanism that warrants discussion is mortality salience. Given the association between mask-wearing and the COVID-19 pandemi, it is possible that masks serve not only as a barrier to self-expression but also as a reminder of illness and vulnerability (see Kellaris et la., 2020 for other correlates). Mortality salience has been shown to influence consumer behavior, often driving individuals toward products that bolster identity (Kasser & Sheldon, 2000). From this perspective, the observed preference for conspicuous products may reflect an attempt to reinforce self-worth or cultural identity in response to existential threat, rather than simply compensating for a lack of self-expression. While the authors focus on compensatory consumption as the primary mechanism, incorporating mortality salience into the theoretical discussion could provide a stronger interpretation of the findings. Given that, as it stands, one could plausibly argue that mortality salience is helping to drive this effect.

As it stands, the studies read a bit demandy in that the authors ask participants to wear a mask and then ask them about their self expression preferences. It seems reasonable that, at least in some cases, people could guess their hypothesis and thus respond accordingly. If authors wish to include additional studies, I’d highly recommend a more subtle measure. I’d also suggest the authors measure the mediator after the DV to account for demand effects.

One last note I think needs to be addressed at some point in this manuscript is the scope of the effect.

  • Is this effect restricted to China, or do you expect it to generalize? If the latter, a limitation would be it’s sample.
  • Does this apply to other forms of attire such as a burqa? Sunglasses? Might be worth mentioning as a future direction.

One of the most admirable aspects of this paper is that it refrains from overstating, the experiments are clean, and the conclusions are straightforward. That being said, identifying the scope of your effect only strengthens the value of your findings.

Minor notes:

  • There were several instances of switching between behavioral and behavioural. I’d recommend picking one spelling and remain consistent. I think Behavioral Sciences prefers the American spelling “behavior” over the British spelling “behaviour.”
  • The three definitions given for compensatory consumption seemed a bit at odds. I’d recommend picking one and sticking to it.
  • The telephone survey in Guangzhou, in my opinion, hurts the paper as it seems to unnecessarily narrow the scope of the paper. I’d recommend removing it, or replacing it with a larger survey.
  • Typo on line 54, “Individuals may find themselves in a unmet due to the restricted avenues for self-54 expression imposed by the mask.”

Potentially helpful citations:

Sittenthaler, S., Traut-Mattausch, E., & Jonas, E. (2015). Observing the restriction of another person: Vicarious reactance and the role of self-construal and culture. Frontiers in psychology6, 1052.

Kellaris, J. J., Machleit, K., & Gaffney, D. R. (2020). Sign evaluation and compliance under mortality salience: lessons from a pandemic. Interdisciplinary Journal of Signage and Wayfinding4(2), 51-66.

Kasser, T., & Sheldon, K. M. (2000). Of wealth and death: Materialism, mortality salience, and consumption behavior. Psychological science11(4), 348-351.

Author Response

Comment 1: The paper looks at an interesting consequence of mask-wearing in the post-pandemic world. Specifically, how mask-wearing, normalized during the COVID-19 pandemic, influences consumers’ need for self-expression and subsequent preferences for conspicuous consumption. Drawing on compensatory consumption processes, the authors propose that mask-wearing limits typical self-expressive outlets (e.g., facial cues), thereby increasing the need for alternative forms of expression through product choices. Across three studies, they demonstrate that mask-wearing heightens the desire for self-expression, which in turn increases preferences for conspicuous products (e.g., items with larger brand logos). This effect appears contingent upon a consumer’s self-construal (i.e., independent vs. interdependent), such that those with independent self-construals show heightened conspicuous self-expression preferences. The paper situates itself nicely within the literature on consumer identity and self-expression, and aims to offer insights into how individual differences shape consumer behavior in the post-pandemic era.

Response 1: Thank you very much for recognizing the interesting focus of our paper. We have taken all your feedback as the guiding support for revising this manuscript. Next is our detailed response to each of your suggestions.

Comment 2: One thing I struggled with in reading your paper lies in the leap from self-expression to conspicuous consumption, particularly when considered through the lens of self-construal. The authors argue that individuals with an independent self-construal, who prioritize autonomy and distinctiveness, are more likely to engage in conspicuous consumption when mask-wearing restricts facial self-expression. However, it is unclear why conspicuous products, such as a Nike sweatshirt with a large logo, would serve as an effective outlet for individual self-expression, given that such items are often mass-produced and socially conforming rather than unique or identity-revealing. This raises the question of whether the observed behavior reflects a compensatory response to a deficit in self-expression, or instead a reactive response to a perceived restriction on autonomy (e.g., reactance). Independent self-construals are more sensitive to constraints, which makes them more susceptible to psychological reactance (Sittenthaler et al., 2015). From this perspective, the preference for conspicuous products may reflect an effort to reassert agency or visibility in the face of externally imposed limitations, rather than to compensate for an expressive deficit. A clearer articulation of why conspicuousness, rather than uniqueness, creativity, or self-brand congruence, serves as the primary expressive outlet would strengthen the results and takeaways from the paper. Additionally, considering or ruling out reactance as an alternative explanation would help clarify the underlying motivational mechanism, particularly given that the self-construal moderation pattern may align more closely with a reactance framework.

Response 2: Thank you for prompting us to examine the logic behind our hypothesis using the moderating role of self-construal as a starting point.

In fact, self-construal moderates the relationship between facial occlusion and the need for self-expression. Individuals with an independent self-construal habitually pay more attention to the self. When faces are occluded, individuals with an independent self-construal experience a perceived significantly diminished social visibility, creating the need to express themselves. In contrast, individuals with an interdependent self-construal rely on shared norms and relational networks rather than fixed personal attributes to maintain their social status. As a result, facial occlusion poses less of a threat to their social identity. Given that facial occlusion induces self-expression needs of varying intensities across self-construal types, independent self-construal individuals may experience more pronounced needs than interdependent ones. In conjunction with Hypothesis 2 (H2), we then formulate our hypothesis 3. We have reconstructed "2.3. Moderating Role of Self-Construal" to make this line of reasoning clearer.

We also have addressed the issue about why conspicuous products serve as an effective channel for individuals' self-expression in our research, by rewriting "2.2. Facial Occlusion: From Self-Expression Needs to Conspicuous Preferences" and the second paragraph of "7.1. Theoretical Implications".

In the section "2. Literature Review and Hypothesis Development," we emphasizes that high public visibility and symbolic standardization are crucial characteristics linking self-expression needs to conspicuous preferences.

Firstly, conspicuous consumption is inherently public in nature; its signaling capacity relies on being observed by others. This not only involves the purchase of high-end products but also includes a preference for eye-catching logos. For individuals with an independent self-construal, when facial self-expression is restricted, the need to be perceived seen by others becomes more pronounced. Conspicuous products can effectively attract the attention of others, thereby satisfying this need. In this regard, self-brand congruence focuses more on the alignment between oneself and brand values or propositions, which is an internal form of identification. This concept itself does not emphasize the outward transmission of signals. However, products characterized by uniqueness and creativity may seem to also fulfill the need for being noticed to some extent. This is where the second characteristic becomes crucial.

Secondly, the symbols employed in conspicuous products are standardized, meaning they follow a set pattern that is widely recognized and understood across various social contexts. As a result, observers require minimal cognitive effort to decode the messages they convey. For example, a well-known brand's logo on a product can instantly communicate a consistent and stable sense of wealth, status, or a particular lifestyle to those who see it. In contrast, unique and creative products often fall short in this regard. Their meanings are highly subjective and lack widespread and standardized recognition. Different individuals, influenced by their backgrounds, experiences, and personal tastes, may interpret the same product in different ways.

From the perspective of transmitting consistent signals outward in the social environment, this standardized and easily interpretable nature of conspicuous product symbols makes them a more reliable and effective means for individuals to communicate their self-image to a broad audience in the social environment, when traditional facial self-expression is limited.

To help readers better understand our logic, in the second paragraph of Section "7.1 Theoretical Implications," we have discussed how to distinguish conspicuous consumption from other potential compensation strategies.

Moreover, regarding alternative explanations for psychological reactance, we have conducted a new experiment, "5.1. Experiment 3," to rule out potential interferences such as psychological reactance that might be induced by the forced manipulation of mask-wearing through random assignment. We also accounted for factors like risk perception that could be triggered by the potential association of face masks with diseases or death. Furthermore, in the second paragraph of "5.3. Discussion," we discussed the possible reasons behind these results. The results show that the social environment in which we conducted the research has a high level of acceptance towards the daily use of masks. This widespread social recognition means that the act of wearing a mask in our experimental setting is not severely perceived as a restriction on freedom that would significantly trigger psychological reactance.

On one hand, this result has allowed us to effectively strip away the influence of psychological reactance when validating our theoretical framework. We can be less concerned about the potential impact like “independent self-construal individuals being more sensitive to restrictions and prone to psychological reactance,” which might have on our research framework, as the social context minimized the likelihood of such a reaction. On the other hand, we fully understand your insight. It is undeniable that this characteristic of our research environment has inevitably limited the applicability scope of our research conclusions. The specific social acceptance of mask-wearing in our study area may not be representative of all regions and situations. Therefore, as we have indicated in the second paragraph of "5.3. Discussion," this effect can be applied to regions with a socio-cultural environment similar to that of China.

We are grateful for your constructive comment. we believe that these revisions have effectively addressed your concerns.

Comment 3: Another potential mechanism that warrants discussion is mortality salience. Given the association between mask-wearing and the COVID-19 pandemic, it is possible that masks serve not only as a barrier to self-expression but also as a reminder of illness and vulnerability (see Kellaris et la., 2020 for other correlates). Mortality salience has been shown to influence consumer behavior, often driving individuals toward products that bolster identity (Kasser & Sheldon, 2000). From this perspective, the observed preference for conspicuous products may reflect an attempt to reinforce self-worth or cultural identity in response to existential threat, rather than simply compensating for a lack of self-expression. While the authors focus on compensatory consumption as the primary mechanism, incorporating mortality salience into the theoretical discussion could provide a stronger interpretation of the findings. Given that, as it stands, one could plausibly argue that mortality salience is helping to drive this effect.

Response 3: Thank you for bringing up the important concept and its potential influence on our findings. We have already taken steps to address this concern.

In "5.1. Experiment 3," we ruled out potential interferences such as risk perception that could be triggered by the potential association of face masks with diseases or death. This helps to isolate the effect of facial occlusion on self-expression needs and subsequent conspicuous preference s from the influence of mortality-related concerns.

Moreover, in the second paragraph of "5.3. Discussion," we delved into the possible reasons behind the results. We believe that this result stems from the widespread social acceptance of mask use in the post-COVID-19 era. The efficacy of masks has been widely recognized by the public, and the act of wearing masks has transitioned from being a special measure to a daily routine. This has essentially minimized the association with freedom restrictions or disease-related anxiety. Certainly, we have also continued to elaborate on the scope of application.

Once again, we appreciate your thoughtful comments, which have encouraged us to think more deeply about the complexities of our research context.

Comment 4: As it stands, the studies read a bit demandy in that the authors ask participants to wear a mask and then ask them about their self expression preferences. It seems reasonable that, at least in some cases, people could guess their hypothesis and thus respond accordingly. If authors wish to include additional studies, I’d highly recommend a more subtle measure. I’d also suggest the authors measure the mediator after the DV to account for demand effects

Response 4: We sincerely appreciate your specific suggestions.

In response to your concern, we have added "4.1. Experiment 1" to enhance the diversity of our experimental manipulation methods. Experiment 1 is a quasi-experiment conducted offline, employing a design in which participants naturally decide whether to wear face masks. In this experiment, participants' autonomous choices in splitting into the facial occlusion and without facial occlusion groups, without any intervention from the experimenters, significantly minimize the likelihood of participants guessing the experimental purpose and adjusting their responses accordingly. Of course, we also discussed the limitations of this method in "4.3. Discussion".

As you point out about the measurement sequence of variables, this is the procedure we followed during the actual implementation of the experiments. However, we failed to clearly present these details in the original manuscript. We have rectified this issue in the revised version.

Once again, we are grateful for your valuable guidance.

Comment 5: One last note I think needs to be addressed at some point in this manuscript is the scope of the effect.

Is this effect restricted to China, or do you expect it to generalize? If the latter, a limitation would be it’s sample.

Does this apply to other forms of attire such as a burqa? Sunglasses? Might be worth mentioning as a future direction.

One of the most admirable aspects of this paper is that it refrains from overstating, the experiments are clean, and the conclusions are straightforward. That being said, identifying the scope of your effect only strengthens the value of your findings.

Response 5: We sincerely appreciate your positive feedback and thank you for bringing up the issue regarding the effect scope.

we have addressed this issue about the generalization, in the second paragraph of "7.3. Limitations and Directions for Future Research". Given that social norms, values, and behaviors related to mask-wearing and self-expression may vary across different countries or regions, conservatively speaking, the conclusions of our current study can be applied to regions with a socio-cultural environment similar to that of China. Future research should replicate our study in diverse cultural settings to assess the cross-cultural validity of our findings and explore potential cultural moderators that may influence the strength or direction of these relationships.

We have also considered your comment about other forms of facial occlusion. While our study focused on face masks as the form of facial occlusion, we recognize that other forms of attire like burqas or sunglasses can also act as barriers to self-expression and may trigger similar psychological and behavioral responses. Given this, our findings have the potential to be applicable to these alternative forms of facial occlusion. In the third paragraph of "7.3. Limitations and Directions for Future Research," we discuss the necessity of further examining the applicability to other forms of facial occlusion in future studies.

Comment 6: Minor notes:

There were several instances of switching between behavioral and behavioural. I’d recommend picking one spelling and remain consistent. I think Behavioral Sciences prefers the American spelling “behavior” over the British spelling “behaviour.”

Response 6-1: Thank you very much for your expert guidance. We followed your advice, chose the American spelling "behavior" you recommended, and standardized the spelling throughout the manuscript.

The three definitions given for compensatory consumption seemed a bit at odds. I’d recommend picking one and sticking to it.

Response 6-2: Thank you very much for pointing out this issue. In this revision, we check and ensure consistency in the definition of each concept.

The telephone survey in Guangzhou, in my opinion, hurts the paper as it seems to unnecessarily narrow the scope of the paper. I’d recommend removing it, or replacing it with a larger survey.

Response 6-3: Thank you very much for raising this concern. After consideration, we have decided to remove it in the revised manuscript.

Typo on line 54, “Individuals may find themselves in a unmet due to the restricted avenues for self-54 expression imposed by the mask.”

Response 6-4: Thank you very much for discovering this spelling error. We have corrected it and conducted a re-examination of the manuscript.

Comment 7: Potentially helpful citations:

Sittenthaler, S., Traut-Mattausch, E., & Jonas, E. (2015). Observing the restriction of another person: Vicarious reactance and the role of self-construal and culture. Frontiers in psychology, 6, 1052. https://doi.org/10.3389/fpsyg.2015.01052

Kellaris, J. J., Machleit, K., & Gaffney, D. R. (2020). Sign evaluation and compliance under mortality salience: lessons from a pandemic. Interdisciplinary Journal of Signage and Wayfinding, 4(2), 51-66.

Kasser, T., & Sheldon, K. M. (2000). Of wealth and death: Materialism, mortality salience, and consumption behavior. Psychological science, 11(4), 348-351.

Response 7: We sincerely appreciate the literature you have listed and all the specific suggestions you have provided. These citations have provided us with great help.

We sincerely thank you again for your time, effort, and valuable guidance.

Round 2

Reviewer 1 Report

Comments and Suggestions for Authors

Thank you for revising and improving this manuscript. It reads slightly better. However, after reading the introduction and literature review, I am still not convinced by the topic of facial occlusions and self-expression needs in this article. Typically, people do not cover their faces in everyday situations, which might limit the practical implications and the relevance of this research. Also, why force an unnatural situation for this research context? People have varied motivations for covering their faces, and they deliberately cover their faces if they want to (it is already part of their self-expression, such as Muslims with their burqa). I am suggesting that the authors do a preceding study to explore the psychology behind why people cover their faces (e.g., because of the weather) to justify this topic, and later on, justify the variables examined in this research.

The authors cited Zhong et al. (2010) to provide context for facial occlusions and self-expression. However, Zhong et al. (2010) explain that darkness (such as face coverage) induces anonymity (the feeling of not being identified). It is almost the opposite of self-expression. The context and two main constructs (facial occlusion and the need for self-expression) of this study require well-justified justification with appropriate citations. The authors should also discuss the psychological reactance and perceived risks in justifying facial occlusions.

The relationship between the variables examined in this research appears somewhat unclear. For instance, in section 2.3, the connection between "individuals with an independent self-construal experiencing perceived diminished social visibility when faces are occluded" and the cited studies, Zhong et al. (2010) on anonymity in darkness, and Morgan & Townsend (2022) on utilitarian vs. hedonic self-expression, needs further clarification. These sources address different issues, and I recommend identifying more appropriate references to support the opinions expressed.

The experimental design needs further justification. Firstly, the experiment context of the mask that is voluntarily worn - since the experiment measures the need for self-expression, the researchers need to justify the willingness to wear a mask amongst the participants, especially in Experiments 2, 3, 4, and 5, where the choice to wear a mask was not up to the individuals. In experiments 1 and 2, it seems the context of wearing a mask is out of nowhere. It is unclear whether the participants have a similar degree of willingness to wear the mask, considering the psychological reactance and perceived risks, before measuring their need for self-expression. In Experiment 4, did the researcher measure the participants' existing level of conspicuous preferences? How to ensure that the measured conspicuous preferences were the result of the facial occlusion situation? In Experiment 5, what did you mean by the demographic information in "Finally, the participants provided demographic information."? Secondly, the participant selection criteria - the criteria of "personal characteristics" are unclear. What kind of personal characteristics were eligible to participate in this study? I noticed that some participants were noted as "high school or below" - were they adult participants? How did the researchers verify whether those participants understood the questionnaire items?

The theoretical and practical contributions of this study appear somewhat limited, particularly given the relatively uncommon and artificial context of facial occlusions in everyday society. Additionally, people often cover their faces voluntarily as a form of self-expression, such as expressing their culture or beliefs. The insights from this research may not apply to these contexts, which further restricts the study’s broader relevance. Considering a context where the researchers can justify the variables with wider real-world application may help strengthen the overall impact of the research.

Overall, my recommendation to the authors is to rewrite this study from a different angle to better justify the concepts and context.

Comments on the Quality of English Language

There are some vague sentences, and the transitions between paragraphs and topics need clarity to improve the information flow.

Author Response

Comment 1: Thank you for revising and improving this manuscript. It reads slightly better. However, after reading the introduction and literature review, I am still not convinced by the topic of facial occlusions and self-expression needs in this article. Typically, people do not cover their faces in everyday situations, which might limit the practical implications and the relevance of this research. Also, why force an unnatural situation for this research context? People have varied motivations for covering their faces, and they deliberately cover their faces if they want to (it is already part of their self-expression, such as Muslims with their burqa). I am suggesting that the authors do a preceding study to explore the psychology behind why people cover their faces (e.g., because of the weather) to justify this topic, and later on, justify the variables examined in this research. 

Response 1: Thank you so much for taking the time to review our manuscript and for acknowledging the improvements we've made. Your insightful comments in both rounds have been invaluable in helping us refine and polish our work.

1.1 Regarding the topic of facial coverings and self-expression needs

We truly appreciate you raising this crucial question. It has prompted us to take a closer look at the gap between the original intention of our research, the essence of our topic, and the logical presented in the introduction and hypothesis development sections. We're grateful for this opportunity to better convey our underlying reasoning to readers.

In fact, our research aims to explore how external conditions can significantly shape these self-expression needs. The human body serves as a vital medium for self-expression, offering a tangible platform for projecting one's inner self. Our starting point was to investigate whether manipulating the body itself (as Embodied cognition theory posits that physical sensations can alter perception, highlighting the body's active role in constructing cognitive experiences) can act as an external stimulus to change a person's level of self-expression. Covering the face is an external physical behavior that directly impacts the body's sensory input. It creates a sensation of enclosure and restriction on the facial skin. Therefore, we infer that when the face is covered, people may feel as though a part of their "self" is being constrained, leading to an imbalance with their current level of self-expression needs. So, under facial covering, the need for self-expression can be surfaced.

We've highlighted the modifications in Section 1. Introduction and Section 2. Literature Review and Hypotheses Development in yellow for your easy reference. We believe that the revisions will address your concerns about the topic of facial occlusions and self-expression needs.

1.2 Regarding the concern about the non-daily nature of facial covering limiting practical implications and relevance

We understand that your concern is reasonable. In response, we have added relevant content in Section 3.2. Facial Covering Stimulus Selection. Although we did not conduct a separate survey to investigate the reasons for mask-wearing, we have delved deeper into the existing literature. While the COVID-19 pandemic significantly accelerated the global adoption of mask, it's important to note that this practice was already customary in at least some Asian regions prior to the pandemic. This situation makes us more cautious about the general applicability of our research conclusions, but it also clarifies the positioning of our study and the validity of choosing Chinese samples. Our research findings are at least reasonable and applicable in the Asian context, and we believe they still hold significant value.

Zhu et al. (2024) further clarified through in-depth interviews that in the post-pandemic context, functional needs—such as medical defense, protection against fog and dust, warmth and moisture retention, and food and beverage hygiene—as well as emotional needs, including safeguarding facial privacy and expressing unique personalities, collectively contribute to the daily use of face masks. It should be emphasized that the core of this study is to examine the impact of facial coverings, which is grounded in the real-world perception of physical occlusion. Consequently, the reasons behind individuals' decisions to cover their faces are not the focal point of our investigation.

On the other hand, in the third paragraph of Section 7.3 "Limitations and Directions for Future Research", we specifically discuss the applicability of transitioning from face masks to other types of coverings. We disclose the following points: "While our study focused on face masks as the form of facial covering, we think that other types of attire, such as burqas, may also induce a sense of physical restriction on the facial skin, potentially triggering similar psychological and behavioral responses. However, different coverings inherently carry distinct connotations and cover different body parts. This may lead to variations in their effects." And "Moreover, conspicuous preferences represent merely a self-expression pathway within the consumption realm, which may be achieved through other means in different contexts."

Once again, we appreciate your comments to guide the rigor of our manuscript.

Comment 2: The authors cited Zhong et al. (2010) to provide context for facial occlusions and self-expression. However, Zhong et al. (2010) explain that darkness (such as face coverage) induces anonymity (the feeling of not being identified). It is almost the opposite of self-expression. The context and two main constructs (facial occlusion and the need for self-expression) of this study require well-justified justification with appropriate citations. The authors should also discuss the psychological reactance and perceived risks in justifying facial occlusions. 

Response 2: Thank you for your assistance in ensuring the rigor of our literature citations. 

Indeed, Zhong et al. (2010) demonstrated that the illusory sense of anonymity induced by sunglasses does not stem from the objective reduction in others' ability to recognize the wearer due to darkness. Instead, it arises from the wearer's subjective, phenomenological experience of darkness. This finding reveals that the body's sensory experiences can directly influence how individuals self-perceive and regulate their behavior. We cited Zhong et al. (2010) to provide context for facial covering and self-expression to support the point that covering the face is an external physical behavior that directly impacts the body's sensory input. The covering creates a sensation of enclosure and restriction on the facial skin, which brings about perceptual impacts on individuals. We have elaborated on this logic in greater detail in the introduction section of the revised manuscript.

We believe there is a clear distinction between the two studies. And our research is not in conflict with Zhong et al. (2010). Zhong et al. (2010) zeroes in on the eyes and how the altered visual input from them leads to a feeling of anonymity. In contrast, our study places much greater emphasis on the face as a holistic entity in self-presentation. Although both are related to the body's sensory experiences, they operate on different psychological levels and have distinct impacts on behavior.

Although the core of this study is to examine the impact of facial coverings, which is grounded in the real-world perception of physical occlusion, we have taken your suggestions into account. In Experiment 3, we discussed potential interferences such as psychological reactance that might be induced by the forced manipulation of mask-wearing through random assignment, and factors like risk perception that could be triggered by the potential association of face masks with diseases or death. And in Section 5.3 Discussion, We state, "In Experiment 3, we took into account potential confounding factors, including psychological reactance and risk perception. Our results show that the mask-wearing manipulation may not have substantially activated notable cognitive differences related to these factors. Given that social norms and values related to mask-wearing may vary across different countries or regions, we circumspectly state that this conclusion can be applied to regions with a socio-cultural environment similar to that of China." This part not only acknowledges the consideration of reactance and perceived risks, but also highlights the cultural-context-dependent nature of our findings. Additionally, in Section 3.3. Sample Acquisition and Ethical Safeguards, we emphasized that all participants were Chinese living in China to ensure the consistency of norms and values that might influence perceptions and behaviors related to mask wearings, which also helps to minimize the impact of these potential confounding factors.

We hope that our explanations and revisions have addressed your concerns.

Comment 3: The relationship between the variables examined in this research appears somewhat unclear. For instance, in section 2.3, the connection between "individuals with an independent self-construal experiencing perceived diminished social visibility when faces are occluded" and the cited studies, Zhong et al. (2010) on anonymity in darkness, and Morgan & Townsend (2022) on utilitarian vs. hedonic self-expression, needs further clarification. These sources address different issues, and I recommend identifying more appropriate references to support the opinions expressed. 

Response 3: Thank you for your suggestions regarding the clarification of variable relationships and the appropriateness of the cited literature. 

In addition to the clarifications and modifications we made in response to Comment 2 concerning the Zhong et al. (2010) literature, we have also revised Section 2.3, "Moderating Role of Self-Construal," as indicated by the highlighted text in the main body of the paper. In essence, we have approached the moderating role of self-construal from a more fundamental perspective. When facial covering occurs, it directly disrupts the established self-expression patterns that individuals are accustomed to. However, the magnitude of the resulting self-expression need gap varies between independent and interdependent self-construal individuals because of their distinct baseline levels of self-expression needs. We hypothesize that independent self-construal individuals, with their inherently higher baseline self-expression needs, will experience a more pronounced self-expression need gap after facial covering. Furthermore, we have been more cautious in citing Morgan & Townsend (2022). We now only use this reference to support the notion that the need for self-expression is a fundamental requirement for creating and maintaining one's self-identity.

Thank you again for your suggestion.

Comment 4: The experimental design needs further justification. Firstly, the experiment context of the mask that is voluntarily worn - since the experiment measures the need for self-expression, the researchers need to justify the willingness to wear a mask amongst the participants, especially in Experiments 2, 3, 4, and 5, where the choice to wear a mask was not up to the individuals. In experiments 1 and 2, it seems the context of wearing a mask is out of nowhere. It is unclear whether the participants have a similar degree of willingness to wear the mask, considering the psychological reactance and perceived risks, before measuring their need for self-expression. In Experiment 4, did the researcher measure the participants' existing level of conspicuous preferences? How to ensure that the measured conspicuous preferences were the result of the facial occlusion situation? In Experiment 5, what did you mean by the demographic information in "Finally, the participants provided demographic information."? Secondly, the participant selection criteria - the criteria of "personal characteristics" are unclear. What kind of personal characteristics were eligible to participate in this study? I noticed that some participants were noted as "high school or below" - were they adult participants? How did the researchers verify whether those participants understood the questionnaire items? 

Response 4: We appreciate your attention to the details of our experimental design. Your comments are valuable to help us present the experimental process in a more transparent manner. Below, we will address your concerns one by one.

4.1 Regarding the Mask Wearing Context

We have provided a more detailed explanation of the differences in mask manipulation across different experiments in the first paragraph of Section 3.3. Sample Acquisition and Ethical Safeguards (highlighted text). In the questionnaire, we clearly stated that participants had the right to choose whether to accept this experimental manipulation, and they could exit directly if they couldn't accept the experimental conditions.

We did not measure the willingness to wear a mask for the following reasons. On one hand, the core of our research is that when the face is covered, it acts as a physical barrier, producing direct and immediate external stimuli. The act of wearing a mask itself can induce psychological changes. Thus, the willingness to wear a mask and the reasons behind individuals' decisions to cover their faces are not the focal points of our investigation. Additionally, since psychological reactance and perceived risk are not the main focuses of our research, we only discussed them in Experiment 3. On the other hand, our experimental manipulations include both quasi-experiments where participants naturally choose whether to wear masks and forced manipulations of wearing or not wearing masks. Especially Experiment 1 (self selection) and Experiment 3 (forced randomization) were conducted in similar offline environments. The results from different manipulations have all verified our hypotheses, which to some extent alleviates your concern.

We agree your concern. Given the generally high acceptance of masks in China, we may downplay the impact of this variable. To expand the applicability of our conclusions in future studies, this limitation needs to be considered. Therefore, we have disclosed this limitation in the second paragraph of Section 7.3. Limitations and Directions for Future Research (highlighted text). We state, “One limitation is the potential restriction of sample generalizability. The research was conducted within a Chinese cultural context. Given that social norms, values, and behaviors related to mask-wearing and self-expression may vary across different countries or regions, the conclusions of our current study can be applied to regions with a socio-cultural environment similar to that of China. For instance, since the common acceptance of face masks in China, we did not specifically take into account the impact of individuals' willingness to wear masks in our research design. Additionally, although psychological resistance didn't seem to significantly affect our research results in this study, in regions where mask-wearing is less socially accepted or in more individualistic cultures, its influence could be stronger (Jonas et al., 2009). This may alter the underlying mechanism or affect result replication. Future research should replicate our study in diverse cultural settings to assess the cross-cultural validity of our findings and explore potential variables that may influence the strength or direction of these relationships.”

4.2 Regarding the Measurement of Conspicuous Preferences

Since Experiment 4 and Experiment 5 were conducted using random sampling and random grouping on an online platform, it to some extent balanced the possibility of significant differences in the original conspicuous preferences between the two groups of participants. Moreover, before implementing the experiments, we considered whether a pre-test was necessary. We believed that a pre-test might lead participants to speculate about the experimental intent, which could affect the measurement results of conspicuous preferences after the manipulation. Therefore, we did not measure the original preferences.

4.3 Regarding the Presentation of Demographic Information

It should be noted that the types of demographic information measured in the online experiments were the same. To ensure rigor, we have revised the previous description in Experiment 5.

4.4 Regarding Participant Selection Criteria

We have provided a more detailed explanation in the second paragraph of Section 3.3. Sample Acquisition and Ethical Safeguards. Participants were required to be at least 18 years old to ensure adequate cognitive and decision-making capacity, and all were Chinese living in China to ensure the consistency of norms and values that might influence perceptions and behaviors related to mask wearings. At recruitment, we excluded participants with several types of medical conditions. Those with respiratory diseases like asthma or bronchitis were not included because these conditions can cause breathing difficulties, and wearing a mask may exacerbate discomfort, distracting participants from experimental tasks and introducing confounding factors unrelated to the embodied effects of facial covering. Participants with facial skin diseases were also excluded.

4.5 Regarding the Concern about Educated Level

We have corrected it to a more understandable writing style and highlighted the relevant parts in each experiment. For example, "this experiment recruited 120 participants (46 males and 74 females, Mage = 31.96, SD = 8.55, rang = 18 - 55) from Credamo.com. Regarding the last finished level of education, 29 had high school education or lower, 72 had a Bachelor's degree, and 19 had a master's degree or higher."

Thanks again. These modifications enhance the rigor of our research.

Comment 5: The theoretical and practical contributions of this study appear somewhat limited, particularly given the relatively uncommon and artificial context of facial occlusions in everyday society. Additionally, people often cover their faces voluntarily as a form of self-expression, such as expressing their culture or beliefs. The insights from this research may not apply to these contexts, which further restricts the study’s broader relevance. Considering a context where the researchers can justify the variables with wider real-world application may help strengthen the overall impact of the research. 

Response 5: Thank you for your attention to the contributions of our study.  

Regarding your concern about the limited contributions due to the relatively uncommon and artificial context of facial occlusions in daily life, as we mentioned in the response to Comment 1, relevant content has been added in Section 3.2 "Facial Covering Stimulus Selection". Although the COVID-19 pandemic sped up the global use of masks, it's worth noting that mask-wearing was already a common practice in some Asian regions before the pandemic. So, our research findings are at least reasonable and applicable in the Asian context, and we believe they hold significant value.

In this revision, we've built our research framework on the embodied cognition theory. Covering the face is an external physical act that directly affects the body's sensory input. Our study focuses on the fact of facial covering and its impact on conspicuous consumption preferences. You pointed out that people cover their faces may as a form of self-expression. This gives us a new angle. We admit there are other ways of self-expression, such as using mask-wearing itself as you mentioned. So, we don't think this conflicts with our research. We've included your point in Section 7.3 "Limitations and Directions for Future Research" of the third paragraph. We state, "conspicuous preferences represent merely a self-expression pathway within the consumption realm, which may be achieved through other means in different contexts." It will inspire future research directions.

Once again, we appreciate your comments and suggestions.

Comment 6: Overall, my recommendation to the authors is to rewrite this study from a different angle to better justify the concepts and context.

Response 6: We appreciate your comments throughout the review process. Following your suggestions above, the revised manuscript has more clearly defined the research positioning and the key logical threads of our deductions. We have made revisions and added necessary details in various sections, including the title, abstract, citations, hypothesis derivation, and methodological descriptions. 

In particular, we would like to take this opportunity to elaborate on the conceptual interpretation angle we adopted in this revision.

We started by affirming that the need for self-expression is a fundamental human drive. While individuals vary in the intensity of their intrinsic motivations, external conditions can also significantly influence these needs. This led us to consider whether manipulating the body itself could serve as an external stimulus to alter a person's level of self-expression. Grounded in the embodied cognition theory, when the face is covered, it directly impacts the body's sensory input. The sensation of enclosure on the face triggers a psychological gap between the current level of self-expression needs and the reduction of available self-expression pathways. By focusing on this external, physical manipulation of the face, we were able to minimize the interference of intrinsic motivations in our research framework. It enhances the reliability of our proposed relationship between facial covering and self-expression needs. Subsequently, we explored the way of substitutive self-presentation, wherein individuals reallocate their expressive efforts to alternative domains when original self-expression pathways are restricted.

Finally, we sincerely appreciate your professional insights and constructive feedback throughout two rounds of the review process. Although limitations in our research scope, experimental materials, and implementation constraints prevented us from fully meeting all your expectations, we have made objective statements in the manuscript to avoid overstating.

Overall, your guidance has undoubtedly enhanced the academic rigor of our work, for which we are truly grateful. Guided by the suggestions from the reviewers, we believe this research can serve as an interdisciplinary attempt to bridge behavioral science with consumer behavior research.

Comment 7: Comments on the Quality of English Language

There are some vague sentences, and the transitions between paragraphs and topics need clarity to improve the information flow. 

Response 7: Your feedback is helpful. We've revised the text, rephrasing ambiguous sentences for clarity and adding transitional phrases to improve the flow between paragraphs and topics. We hope these changes address the concerns.

Reviewer 3 Report

Comments and Suggestions for Authors

Overall

I appreciate the authors' thorough and thoughtful response to the initial review. They made significant improvements to the manuscript across three main areas: (1) strengthening the theoretical rationale, (2) addressing alternative explanations, and (3) clarifying methodological procedures. In my view, each of these areas is now considerably stronger.

The main remaining concern is the writing itself. The revised manuscript feels harder to follow, and the narrative flow has become somewhat choppy and uneven. Several sections now feel forced, particularly where the authors are working to integrate additional theoretical distinctions or justifications. These issues don’t detract from the theoretical value or empirical contributions of the paper, but they may affect its readability and clarity. A focused round of copyediting and structural tightening would greatly enhance the manuscript's impact.

Intro

I’m not entirely convinced by the term “occlusion.” While technically accurate, it may be less accessible to broader audiences, particularly those outside of academia. Given that citations will likely be driven by the broader implications of face coverings on consumption, I’d encourage the authors to consider a more intuitive term like “facial covering” or “masking.” That said, I leave it to the authors to decide, but this may be worth revisiting from a positioning perspective.

Experiment 1

It’s intriguing that participants who chose to wear a mask also reported a higher need for self-expression. On the surface, this feels counterintuitive… why would those who voluntarily occlude their face feel more compelled to express themselves? One possible psychological bridge (which the authors may wish to elaborate on) is that mask-wearers may exert more control over how they express themselves, shifting that effort to visible, controllable domains like clothing: “Look at what I’m wearing, not my face.” This interpretation aligns with the line in the intro: “Under occlusion, material symbols (e.g., clothing, accessories) provide a controllable and visible alternative.”

Still, I find this study more confusing than clarifying. Given the later clarification that mediators were measured after the DV in other experiments, I’m not sure Experiment 1 adds much. It seems to raise more questions than it resolves.

Also, the appropriate citations for measuring the mediator after the dependent variable as a way to reduce demand effects are:

Kardes, F. R., & Herr, P. M. (2019). Experimental research methods in consumer psychology. In Handbook of research methods in consumer psychology (pp. 3–16). Routledge.

Kardes, F., Fischer, E., Spiller, S., Labroo, A., Bublitz, M., Peracchio, L., & Huber, J. (2022). Commentaries on ‘An Intervention-Based Abductive Approach to Generating Testable Theory,’. Journal of Consumer Psychology32(1), 194-207.

I recommend citing at least one of these sources to support your methodological approach, though the choice is up to the authors.

Experiment 3

I wasn’t able to locate the specific reactance items used in the cited paper. The measure also reads more like an individual difference variable rather than a state-level construct, which was the intention behind my original comment (see Sittenthaler et al., 2015). You might clarify or revise how reactance was operationalized here.

Also, I’d advise against making definitive claims based on null results. For instance, the statement “ruled out interference from these confounding variables” (line 298) overstates the evidence. A more cautious interpretation would be preferable.

Additionally, Experiment 3 appears to show partial mediation, which I think is both reasonable and realistic, but please specify this explicitly in the text.

On a related note, please revise the mediation figure to reflect the fact that all variables were measured, not manipulated (e.g., use rectangles for all constructs). This applies to all figures in the manuscript.

Experiment 4

Please clarify that the test reported is the direct effect of facial occlusion on conspicuous preferences (Line 377). The current language is hard to parse and briefly had me thinking you were reporting a regression or model without the mediator.

The finding that the direct effect was numerically larger but nonsignificant is interesting and deserves a clearer summary.

If you used Hayes’ PROCESS model for the mediation, be sure to cite it:

Hayes, A. F. (2017). Introduction to mediation, moderation, and conditional process analysis: A regression-based approach. Guilford Press.

Discussion

I recommend citing Jonas et al. (2009) again in the discussion and noting that, although reactance did not appear to influence the results in your studies, this may not hold in other cultural contexts. In settings where mask-wearing is less socially accepted, or in more individualistic cultures, the connection between independence and reactance may be stronger. This could shift the underlying mechanism and influence whether the effect replicates. Framing this as a boundary condition not only clarifies the scope of your findings but also provides a useful direction for future research.

---

Overall, the authors did an excellent job addressing my concerns, and I appreciate the care they took in revising the manuscript. I applaud their efforts and wish them the best moving forward.

Author Response

Comment 1: Overall

I appreciate the authors' thorough and thoughtful response to the initial review. They made significant improvements to the manuscript across three main areas: (1) strengthening the theoretical rationale, (2) addressing alternative explanations, and (3) clarifying methodological procedures. In my view, each of these areas is now considerably stronger. 

The main remaining concern is the writing itself. The revised manuscript feels harder to follow, and the narrative flow has become somewhat choppy and uneven. Several sections now feel forced, particularly where the authors are working to integrate additional theoretical distinctions or justifications. These issues don’t detract from the theoretical value or empirical contributions of the paper, but they may affect its readability and clarity. A focused round of copyediting and structural tightening would greatly enhance the manuscript's impact. 

Response 1: Thank you so much for your assessment of our efforts to improve the manuscript. We truly appreciate your recognition of these enhancements. To further enhance the manuscript's readability and the tightness of its logical structure, we've made additional efforts.

Before responding point-by-point to your comments below, we'd like to take this opportunity to elaborate on the conceptual interpretation angle we adopted in this revision.

We started by affirming that the need for self-expression is a fundamental human drive. While individuals vary in the intensity of their intrinsic motivations, external conditions can also significantly influence these needs. This led us to consider whether manipulating the body itself could serve as an external stimulus to alter a person's level of self-expression. Grounded in the embodied cognition theory, when the face is covered, it directly impacts the body's sensory input. The sensation of enclosure on the face triggers a psychological gap between the current level of self-expression needs and the reduction of available self-expression pathways. By focusing on this external, physical manipulation of the face, we were able to minimize the interference of intrinsic motivations in our research framework. It enhances the reliability of our proposed relationship between facial covering and self-expression needs. Subsequently, we explored the way of substitutive self-presentation, wherein individuals reallocate their expressive efforts to alternative domains when original self-expression pathways are restricted.

We believe that this round of modifications and responses has significant progress. Next, please refer to the point-to-point response below.

Comment 2: Intro

I’m not entirely convinced by the term “occlusion.” While technically accurate, it may be less accessible to broader audiences, particularly those outside of academia. Given that citations will likely be driven by the broader implications of face coverings on consumption, I’d encourage the authors to consider a more intuitive term like “facial covering” or “masking.” That said, I leave it to the authors to decide, but this may be worth revisiting from a positioning perspective. 

Response 2: We truly appreciate your consideration of the manuscript's accessibility to a broader audience. We completely agree with your point that while “occlusion” is technically accurate, it might not be as intuitive for non-academic readers. We have adopted the term “facial covering” throughout the revised manuscript.

Once again, we are grateful for your suggestion.

Comment 3: Experiment 1

It’s intriguing that participants who chose to wear a mask also reported a higher need for self-expression. On the surface, this feels counterintuitive… why would those who voluntarily occlude their face feel more compelled to express themselves? One possible psychological bridge (which the authors may wish to elaborate on) is that mask-wearers may exert more control over how they express themselves, shifting that effort to visible, controllable domains like clothing: “Look at what I’m wearing, not my face.” This interpretation aligns with the line in the intro: “Under occlusion, material symbols (e.g., clothing, accessories) provide a controllable and visible alternative.” 

Still, I find this study more confusing than clarifying. Given the later clarification that mediators were measured after the DV in other experiments, I’m not sure Experiment 1 adds much. It seems to raise more questions than it resolves.

Also, the appropriate citations for measuring the mediator after the dependent variable as a way to reduce demand effects are:

Kardes, F. R., & Herr, P. M. (2019). Experimental research methods in consumer psychology. In Handbook of research methods in consumer psychology (pp. 3–16). Routledge.

Kardes, F., Fischer, E., Spiller, S., Labroo, A., Bublitz, M., Peracchio, L., & Huber, J. (2022). Commentaries on ‘An Intervention-Based Abductive Approach to Generating Testable Theory,’. Journal of Consumer Psychology, 32(1), 194-207.

I recommend citing at least one of these sources to support your methodological approach, though the choice is up to the authors.

Response 3: Thank you so much for your guidance on enhancing the experimental rigor. 

We'd like to clarify the design of Experiment 1 and Experiment 2. These experiments were designed to examine the relationship between facial covering and mediator (need for self-expression). Therefore, the dependent variable (conspicuous preferences) was not measured. We further elaborate on this in the yellow text of experiment 1, such as “Given that this experiment measures only a few variables, to lower the chance of participants inferring the true experimental objective, each participant was instructed to mentally simulate a hypothetical shopping scenario. Next, participants completed a paper-based questionnaire to assess their need for self-expression.”

In response to the seemingly counterintuitive finding that participants who chose to wear masks reported a stronger need for self-expression, we have now more clearly elucidated the underlying mechanism through a comprehensive refinement of our logical framework. Under facial covering, the surfaced need for self-expression stems from the fact that facial covering creates a gap between an individual's current level of self-expression needs and the reduction of self-expression pathways. As a result, individuals may feel a stronger urge to express themselves through other visible and controllable means, like clothing choices, as a way to compensate for the loss of facial self-expression. This logical pathway has been explicitly incorporated into the revised manuscript to enhance readers' understanding of our research.

We are also grateful for the reference you provided. Given that Experiment 1 and Experiment 2 did not involve the measurement of a DV, we have incorporated the recommended literature into Experiment 3, which investigates the positive impact of facial covering on conspicuous preferences and verifies the mediating role of the need for self-expression in this relationship.

Once again, we sincerely thank you for your comments, which have significantly improved the quality and clarity of our study.

Comment 4: Experiment 3

I wasn’t able to locate the specific reactance items used in the cited paper. The measure also reads more like an individual difference variable rather than a state-level construct, which was the intention behind my original comment (see Sittenthaler et al., 2015). You might clarify or revise how reactance was operationalized here. 

Also, I’d advise against making definitive claims based on null results. For instance, the statement “ruled out interference from these confounding variables” (line 298) overstates the evidence. A more cautious interpretation would be preferable. 

Additionally, Experiment 3 appears to show partial mediation, which I think is both reasonable and realistic, but please specify this explicitly in the text. 

On a related note, please revise the mediation figure to reflect the fact that all variables were measured, not manipulated (e.g., use rectangles for all constructs). This applies to all figures in the manuscript. 

Response 4: We sincerely appreciate the guidance you've provided on the presentation of our experiments. We've learned a great deal from your comments and have made corresponding revisions as detailed below.

4.1 Psychological Reactance Measurement

Firstly, we'd like to clarify that psychological reactance in our study is a state-level construct, adapted from Jonas et al. (2009). The measurement items you mentioned from Sittenthaler et al. (2015) also originate from the same source article. We have now presented the specific items in the manuscript (see the yellow-highlighted text in the fourth paragraph of 5.1. Experiment 3). The six items (Cronbach's α = 0.881) are as follows: "How reasonable do you think this requirement is?", "How restricted do you feel in your freedom regarding facial covering?", "How much does it bother you?", "How irritated do you probably feel?", "How legitimate is the rule of mandating you to wear (or not wear) a face mask?", and "How much pressure do you feel to comply with this requirement".

4.2 Interpretation of Results

Regarding the interpretation of "The study observed no significant differences in psychological reactance and risk perception between the experimental and control groups", we fully agree with your point on the need for cautious interpretation. In the revised manuscript, we have made revisions in 5.1.2. Results and 5.3. Discussion (highlighted in yellow). For example, “In Experiment 3, we took into account potential confounding factors, including psychological reactance and risk perception. Our results show that the mask-wearing manipulation may not have substantially activated notable cognitive differences related to these factors. Given that social norms and values related to mask-wearing may vary across different countries or regions, we circumspectly state that this conclusion can be applied to regions with a socio-cultural environment similar to that of China.”

4.3 Partial Mediation in Experiment 3

We have specifically discussed the partial mediation result of Experiment 3 in the third paragraph of 5.3. Discussion. The text is as follows: "A particular finding emerges from the differential mediating patterns observed across experimental modalities. In the (offline) Experiment 3, the need for self-expression partly mediated the relationship, while (online) Experiment 4 revealed a full mediation (the direct effect was not significant). It likely reflects the complexity of real-world interactions. Compared to online settings, physical laboratory spaces introduce uncontrolled variables such as ambient noise, incidental social observations, and interpersonal proximity. These factors may activate other psychological or behavioral mechanisms beyond self-expression needs. Experiment 4 provided a more isolated testing environment, which allowed for a clearer examination of the mediating mechanism. However, in Experiment 4, we found that the direct effect was numerically larger than the indirect effect. This suggests that we cannot completely neglect the existence of other paths. Therefore, this study cautiously posits that the need for self-expression plays a partial mediating role in the relationship between facial covering and conspicuous preferences."

4.4 Revision of Mediation Figures

Finally, we are grateful for your guidance on the standard presentation of mediation figures. We have replaced the corresponding mediation figures for Experiment 3 and Experiment 4.

Once again, thank you for your constructive feedback, which have improved the quality of our manuscript.

Comment 5: Experiment 4

Please clarify that the test reported is the direct effect of facial occlusion on conspicuous preferences (Line 377). The current language is hard to parse and briefly had me thinking you were reporting a regression or model without the mediator. 

The finding that the direct effect was numerically larger but nonsignificant is interesting and deserves a clearer summary. 

If you used Hayes’ PROCESS model for the mediation, be sure to cite it:

Hayes, A. F. (2017). Introduction to mediation, moderation, and conditional process analysis: A regression-based approach. Guilford Press.

Response 5: We appreciate your constructive guidance. We have made the following revisions in response to your suggestions:

We have added the citation of Hayes (2017), and have corrected the narrative regarding the result interpretation to make it clearer. The text is as follows: “Then, mediation analysis was conducted using conspicuous preferences as the dependent variable and wearing face masks (0 = without, 1 = with) as the independent variable. The need for self-expression was designated as the mediator, with gender, age and education level included as control variables. The variables were entered into Model 4 of the PROCESS procedure (Hayes, 2017; see Figure 4; coefficients have been standardized). The difference in the need for self-expression between the with-mask-wearing and without-mask-wearing conditions (β = 0.852, p < 0.001, 95% CI = [0.448, 0.956]), and the difference in conspicuous preferences attributable to the difference in the need for self-expression (β = 0.209, p < 0.05, 95% CI = [0.074, 1.028]), were significant and positive.  The results revealed a significant mediation effect of face mask wearing (with vs. Without) on conspicuous preferences (indirect effect (a × b) = 0.178, 95% CI = [0.035, 0.347]) and a significant direct effect (direct effect [c] = 0.287, 95% CI = [-0.160, 1.406]), indicating full mediation by the need for self-expression in this experimental scenario.”

As you pointed out, the finding that the direct effect was numerically larger but non-significant is interesting. In the third paragraph of 5.3. Discussion, we have comprehensively discussed the mediation results from both experiments, taking into account the numerically larger direct effect in Experiment 4. For example, we mentioned that compared to online settings, physical laboratory spaces in Experiment 3 introduce uncontrolled variables such as ambient noise, incidental social observations, and interpersonal proximity, which may activate other psychological or behavioral mechanisms beyond self-expression needs. In contrast, Experiment 4 provided a more isolated testing environment, allowing for a clearer examination of the mediating mechanism. However, the numerically larger direct effect in Experiment 4 suggests that we cannot completely neglect the existence of other paths. Therefore, this study cautiously posits that the need for self-expression plays a partial mediating role in the relationship between facial covering and conspicuous preferences.

We are grateful for your suggestions, which have deepened our interpretation of the results and improved the overall rigor of our research.

Comment 6: Discussion

I recommend citing Jonas et al. (2009) again in the discussion and noting that, although reactance did not appear to influence the results in your studies, this may not hold in other cultural contexts. In settings where mask-wearing is less socially accepted, or in more individualistic cultures, the connection between independence and reactance may be stronger. This could shift the underlying mechanism and influence whether the effect replicates. Framing this as a boundary condition not only clarifies the scope of your findings but also provides a useful direction for future research.

Response 6: We are truly grateful for your suggestions.

In response to your suggestion, we have made the following modifications:

6.1 Section 5.3 Discussion

 We state, "In Experiment 3, we took into account potential confounding factors, including psychological reactance and risk perception. Our results show that the mask-wearing manipulation may not have substantially activated notable cognitive differences related to these factors. Given that social norms and values related to mask-wearing may vary across different countries or regions, we circumspectly state that this conclusion can be applied to regions with a socio-cultural environment similar to that of China." This part not only acknowledges the consideration of reactance but also highlights the cultural-context-dependent nature of our findings.

6.2 Section 7.3. Limitations and Directions for Future Research

We further elaborate on this point. We mention that "although psychological resistance didn't seem to significantly affect our research results in this study, in regions where mask-wearing is less socially accepted or in more individualistic cultures, its influence could be stronger (Jonas et al., 2009). This may alter the underlying mechanism or affect result replication. Future research should replicate our study in diverse cultural settings to assess the cross-cultural validity of our findings and explore potential variables that may influence the strength or direction of these relationships." By doing so, we frame the potential influence of psychological resistance, which helps clarify the scope of our current findings.

Once again, we are deeply grateful for your guidance.

Comment 7: Overall, the authors did an excellent job addressing my concerns, and I appreciate the care they took in revising the manuscript. I applaud their efforts and wish them the best moving forward.

Response 7: We've learned a lot from your suggestions throughout this review process. Your expertise feedback have not only enhanced the quality of our research but also broadened our perspectives on the subject matter. Once again, thank you so much for your time, patience, and invaluable guidance.